# Physics-Constrained Flow Matching: Sampling Generative Models with Hard Constraints

**Utkarsh** *
Massachusetts Institute of Technology

**Pengfei Cai***
Massachusetts Institute of Technology

**Alan Edelman**
Massachusetts Institute of Technology

**Rafael Gomez-Bombarelli** †
Massachusetts Institute of Technology

**Christopher Vincent Rackauckas** †
Massachusetts Institute of Technology

## Abstract

Deep generative models have recently been applied to physical systems governed by partial differential equations (PDEs), offering scalable simulation and uncertainty-aware inference. However, enforcing physical constraints, such as conservation laws (linear and nonlinear) and physical consistencies, remains challenging. Existing methods often rely on soft penalties or architectural biases that fail to guarantee hard constraints. In this work, we propose Physics-Constrained Flow Matching (PCFM), a zero-shot inference framework that enforces *arbitrary nonlinear constraints* in pretrained flow-based generative models. PCFM continuously guides the sampling process through physics-based corrections applied to intermediate solution states, while remaining aligned with the learned flow and satisfying physical constraints. Empirically, PCFM outperforms both unconstrained and constrained baselines on a range of PDEs, including those with shocks, discontinuities, and sharp features, while ensuring exact constraint satisfaction at the final solution. Our method provides a flexible framework for enforcing hard constraints in both scientific and general-purpose generative models, especially in applications where constraint satisfaction is essential.

## 1 Introduction

Deep generative modeling provides a powerful and data-efficient framework for learning complex distributions from finite samples. By estimating an unknown data distribution and enabling sample generation via latent-variable models, deep generative methods have achieved state-of-the-art performance in a wide range of domains, including image synthesis [1–4], natural language generation [5, 6], and applications in molecular modeling and materials simulation [7–9].

Inspired by these successes, researchers have begun applying generative modeling to physical systems governed by Partial Differential Equations (PDEs) [10–13]. In these settings, generative models offer unique advantages, including efficient sampling, uncertainty quantification, and the capacity to model multimodal solution distributions [10, 12]. However, a fundamental challenge in this context is ensuring that generated samples respect the governing physical constraints of the system [12, 14]. In traditional domains like vision or text, domain structure is often incorporated

---

*Equal contribution. Order decided by coin toss.

†Corresponding authors: `rafagb@mit.edu`, `crackauc@mit.edu`

    The code implementations are available in Python and Julia.

39th Conference on Neural Information Processing Systems (NeurIPS 2025).

through *soft constraints*—classifier guidance [3], score conditioning [2], or architectural priors such as equivariance [15]. Manifold-based approaches further constrain generations to lie on known geometric spaces [16–18]. While such methods can align the model with geometric priors, they cannot be easily adapted for enforcing physical laws in dynamical systems.

Crucially, constraint enforcement in generative modeling for PDEs follows a different paradigm. Physical invariants such as mass, momentum, and energy [19, 20] often arise from underlying symmetries [21]. Prior work to incorporate physics into neural networks has largely relied on *inductive biases* in training and regression-based tasks: encoding conservation laws as soft penalties (e.g., Physics-Informed Neural Networks (PINNs)) [22], imposing architectural priors [23, 24]. However, soft constraints can lead to critical failure modes, particularly when exact constraint satisfaction is essential for stability or physical plausibility [25–27]. To address this, recent efforts have explored hard constraint enforcement, through learning conservation laws [28, 29], constraint satisfaction at inference [14, 30], and differentiable physics [31–33].

Despite this progress, hard constraint enforcement in generative models, particularly for PDEs, remains a nascent area [12, 34]. Enforcing hard constraints in generative models is particularly challenging due to the inherent stochasticity of the sampling process, and the constraints must be satisfied exactly in the final denoised solution but need not be preserved throughout the sampling process. DiffusionPDE [10] and D-Flow [35] propose gradient-based constraint enforcement during sampling, but these methods often require backpropagation through expensive PDE operators and may fail to exactly satisfy the target constraints. The ECI framework [12] introduces a novel mixing-based correction process for zero-shot constraint satisfaction, but only empirically evaluates on simple linear constraints and shows limited robustness on sharp, high-gradient PDEs with shocks or discontinuities.

In this work, we introduce **Physics-Constrained Flow Matching (PCFM)**, a framework that bridges modern generative modeling with classical ideas from numerical PDE solvers. PCFM enables *zero-shot constraint enforcement up to machine precision* for pretrained flow matching models, projecting intermediate flow states onto constraint manifolds at inference time, without requiring gradient information during training. Unlike prior methods, PCFM can enforce *arbitrary nonlinear equality constraints*, including global conservation laws, nonlinear residuals, and sharp boundary conditions. It requires no retraining and no architectural modification, operating entirely post-hoc. While we focus on PDE-constrained generation in this work, PCFM provides a flexible framework for enforcing hard constraints for flow-based generative models and may be extended to work for other domains such as molecular design and scientific simulations beyond PDEs. To summarize, we list our contributions as follows:

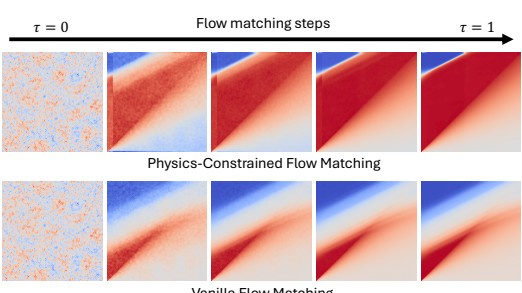

Figure 1: **Evolution of generated solutions for the Burgers equation using vanilla Flow Matching (bottom) and our Physics-Constrained Flow Matching (top).** Burgers' equation exhibits sharp shock fronts (top left in the figure), which standard FFM fails to capture accurately, resulting in overly smoothed or smeared solutions. In contrast, PCFM efficiently incorporates physical constraints during sampling, enabling accurate shock resolution and physically consistent final outputs.

1. We introduce a general framework PCFM for enforcing *arbitrary and multiple physical constraints* in **Flow Matching**-based generative models. These constraints include satisfying conservation laws, boundary conditions, or even arbitrary non-linear constraints. Our method enforces these constraints as hard requirements at inference time, without modifying the underlying training objective.

2. Our approach is *zero-shot*: it operates directly on any pre-trained flow matching model without requiring gradient information for the constraints during training. This makes the method broadly applicable and computationally efficient, especially in scenarios where constraint gradients are expensive or unavailable.

Table 1: Comparison of generation methods motivated by constraint guidance or enforcement.

| | Zero-shot | Continuous Guidance | Constraint Satisfaction | Complex Constraints |
|---|---|---|---|---|
| Conditional FFM [11] | ✗ | ✓ | ✓ | ✗ |
| DiffusionPDE [10] | ✓ | ✗ | ✗ | ✓ |
| D-Flow [35] | ✓ | ✓ | ✗ | ✓ |
| ECI [12] | ✓ | ✓ | ✓ | ✗ |
| PCFM (Ours) | ✓ | ✓ | ✓ | ✓ |

3. We demonstrate significant improvements in generating solutions to partial differential equations (PDEs), outperforming state-of-the-art baselines by up to **99.5%** in standard metrics, such as mean squared error, while ensuring **zero constraint residual**.

4. We evaluate our method on challenging PDEs exhibiting **shocks, discontinuities, and sharp spikes**—settings in which standard flow matching models typically fail. Our approach improves the accuracy of such models *at inference time only*, without the need for retraining or fine-tuning, by retrofitting physical consistency into generated samples.

5. To enable practical deployment, we develop a custom, **batched and differentiable solver** that projects intermediate flow states onto the constraint manifold. This solver integrates seamlessly with modern deep learning pipelines and enables end-to-end differentiability through the constraint enforcement mechanism.

## 2 Related Work

**Flow-based Generative Models.** Flow-based generative models [4, 36, 37] have emerged as a scalable alternative to diffusion models by defining continuous normalizing flows (CNF) using ordinary differential equations (ODEs) [38, 39, 36] parameterized by a time-dependent vector field. In flow matching models, samples from a tractable prior distribution are transported to a target distribution via a learned vector field through a simulation-free training of CNFs. Stochastic interpolants provide a unifying framework to bridge deterministic flows and stochastic diffusions, enabling flexible interpolations between distributions [37]. Furthermore, methods like rectified flows proposed efficient sampling with fewer integration steps by straightening the flow trajectories [36]. Functional flow matching (FFM) [11] and Denoising Diffusion Operator (DDO) [40] extend this paradigm to spatiotemporal data, learning flow models directly over function spaces such as partial differential equations (PDEs) solutions.

**Constraint Guided Generation.** Guiding generative models with structural constraints is an upcoming direction used to improve fidelity in physics-based domains [12, 10, 41]. Constraint information is typically exploited via gradient backpropagation [42, 10, 41, 35] and has been successively applied to domains such as inverse problems [43–45]. Gradient backpropagation through an ODE solver can be prohibitively expensive for functional data such as PDEs [46–48]. Manifold-based flows and diffusion models [16, 18] capture known geometric priors. However, they are not suitable for PDEs having with data-driven or implicit constraints. For PDE-constrained generation, DIF-FUSIONPDE[10] applies PINN-like soft penalties during sampling, while D-FLOW[35] optimizes a noise-conditioned objective. Both approaches incur a high computational cost and offer only approximate constraint satisfaction. In the context of scored-based diffusion models, PDM [49] projects every partially denoised intermediate sample during the sampling phase. However, enforcing constraints at every timestep can over-constrain the trajectory, especially under nonlinear or global constraints like ICs or mass conservation that only apply at the final state. ECI [12] introduces a novel gradient-free, zero-shot, and hard constraint method on PDE solutions. However, its empirical evaluation is limited to simple linear and non-overlapping constraints—e.g., pointwise or regional equalities—with known closed-form projections. It lacks a general roadmap for nonlinear or coupled constraints and has limited evaluation on harder-to-solve PDEs with shocks. Furthermore, while labeled "gradient-free", its reliance on analytical projections restricts extensibility to nonlinear or coupled constraints, and practical enforcement still implicitly relies on nonlinear optimization, which often requires gradient information [50]. We summarize our differences compared to generative methods motivated by constraint satisfaction in Table 1.

**Physics-Informed Learning and Constraint-Aware Numerics.** Physics-informed learning incorporates physical laws as inductive biases in machine learning models, typically through soft penalties as in PINNs [22, 51] or Neural Operators [23, 52, 53]. While effective for regression, these methods lack guarantees of hard constraint satisfaction. Hard constraints have been addressed via

differentiable layers [54, 31, 55, 56], inference-time projections [14, 30], and structured architectures [32, 33, 28]. Constraint-aware integration methods offer complementary insights. constrained neural ODEs [57, 58, 29], differential algebraic equations [59, 60], and geometric integrators [61, 61] enforce feasibility through projection-based updates. Though underexplored in generative modeling, these methods motivate principled approaches to constrained sampling. Our method combines this numerical perspective with flow-based generation, enabling exact constraint enforcement without retraining.

# 3 Methodology and Setup

## 3.1 Problem Setup

We consider physical systems governed by parameterized conservation laws of the form

$$\partial_t u(x,t) + \nabla \cdot \mathcal{F}_\phi(u(x,t)) = 0, \qquad \forall\, x \in \Omega,\ t \in [0,T], \tag{1}$$

$$u(x,0) = \alpha_0(x), \qquad \forall\, x \in \Omega, \tag{2}$$

$$\mathcal{B}u(x,t) = 0, \qquad \forall\, x \in \partial\Omega,\ t \in [0,T], \tag{3}$$

where $\Omega \subset \mathbb{R}^d$ is a bounded domain, $u : \Omega \times [0,T] : \mathcal{X} \to \mathbb{R}^n$ is the solution field, $\mathcal{F}_\phi(u)$ is a flux function parameterized by $\phi \in \Phi$, $\alpha_0$ specifies the initial condition, and $\mathcal{B}$ denotes the boundary operator. For a fixed PDE family and parameter set $\Phi$, we define the associated solution set $\mathcal{U}_\mathcal{F} := \{u \in \mathcal{U} : \exists\, \phi \in \Phi \text{ such that } u \text{ satisfies } (1)-(3)\}$ representing all physically admissible solutions generated by varying $\phi$. We assume access to a pretrained generative model like FFM [11], that approximates this solution set, e.g., via a flow-based model trained on simulated PDE solutions.

In addition to the governing equations, we consider a physical constraint which we wish to enforce on the solution $u(x,t)$ through a constraint operator $\mathcal{H}u(x,t) = 0$ defined on a subdomain $\mathcal{X}_\mathcal{H} \subseteq \Omega \times [0,T]$, and let $\mathcal{U}_\mathcal{H} := \{u \in \mathcal{U} : \mathcal{H}u(x,t) = 0 \text{ for all } (x,t) \in \mathcal{X}_\mathcal{H}\}$ denote the constraint-satisfying solution set. Our objective is to generate samples from the intersection $\mathcal{U}_{\mathcal{F}|\mathcal{H}} := \mathcal{U}_\mathcal{F} \cap \mathcal{U}_\mathcal{H}$, i.e., functions that satisfy both the PDE and the constraint exactly. Importantly, we seek to impose $\mathcal{H}$ at inference time, without retraining the pre-trained model, thus narrowing the generative support from $\mathcal{U}_\mathcal{F}$ to $\mathcal{U}_{\mathcal{F}|\mathcal{H}}$ in a zero-shot manner.

## 3.2 Generative Models via Flow-Based Dynamics

Let $\mathcal{U}$ denote the space of candidate functions $u : \mathcal{X} \to \mathbb{R}^n$, where $\mathcal{X} := \Omega \times [0,T]$ is the spatiotemporal domain of the PDE. We define a family of time-dependent diffeomorphic map $\phi_\tau : \mathcal{U} \times [0,1] \to \mathcal{U}$ called *flow*, indexed by the flow time $\tau \in [0,1]$. A vector field $v_t$ which defines the evolution of the flow $\phi_\tau$ according to the ordinary differential equation: $\frac{d}{d\tau}\phi_\tau(u) = v_\tau(\phi_\tau(u)),\quad \phi_0(u_0) = u_0$. This yields a continuous path of measures through a push-forward map $\pi_\tau = (\phi_\tau)_\# \pi_0$, connecting the prior noise measure $\pi_0$ to the predicted target measure $\pi_1$. Chen et al. [38] proposed to learn the vector field $v_t$ with a deep neural network parameterized by $\theta$. Later in this text, we will denote this parameterized vector field as $v_\theta := v_\tau(x;\theta)$. This naturally induces parameterization for $\phi_t$ called *Continuous Normalizing Flows* (CNFs). Lipman et al. [4] introduced a simulation and likelihood free training method for CNFs called *Flow Matching* along with with other similar works [36, 39] (see Appendix G).

In the context of functional data such as PDE solutions, the *Functional Flow Matching* (FFM) framework [11] extends this idea to functional spaces. Instead of modeling samples in finite-dimensional Euclidean space, FFM learns flows between functions $u_0, u_1 \in \mathcal{U}$ via interpolants [4, 37, 36], trained on trajectories with $u_0 \sim \pi_0$, $u_1 \sim \nu$, where $\nu$ is the target probability measure from where wish to sample the PDE solutions. The vector field $v_\theta$ is parameterized by a Neural Operator [23, 24] which is forward compatible to operate on functional spaces. The resulting generative model approximates the target measure $\pi_1 \approx \nu$ over the support $\mathcal{U}_\mathcal{F}$ through sufficient training (see Appendix G.2 for further details). In this work, we focus on augmenting such pre-trained models to enforce additional physical constraints at inference time, without requiring retraining.

### 3.2.1 Constraint Types in PDE Systems

Constraints in PDE-governed physical systems arise from a variety of sources, including boundary conditions, conservation laws [20, 62], and physical admissibility requirements [63]. These constraints can be local (e.g., pointwise Dirichlet or Neumann conditions) or global (e.g., integral conservation of mass or energy), may be linear or nonlinear in the solution field $u(x,t)$, and may not

Table 2: Summary of constraint types commonly encountered in PDE-based physical systems, categorized by their mathematical form and scope.

| Constraint Type | Representative Form | Linearity |
|---|---|---|
| Dirichlet IC / BC | $\mathcal{H}u = Au - b = 0$ | Linear |
| Global mass conservation (periodic BCs) | $\mathcal{H}u = \int_\Omega u(x,t)\,dx - C = 0$ | Linear |
| Nonlinear conservation law | $\mathcal{H}u = \frac{d}{dt}\int_\Omega \rho(u(x,t))\,dx = 0$ | Nonlinear |
| Neumann or flux boundary condition | $\mathcal{H}u = \partial_n u(x,t) - g(x,t) = 0$ | Potentially nonlinear |
| Coupled or implicit constraints | Nonlinear spatial/temporal relationships | Nonlinear |

be implicitly satisfied by the generated solution. Table 2 summarizes representative forms of commonly encountered constraints, categorized by their mathematical structure. This taxonomy guides the design of appropriate enforcement mechanisms in generative modeling pipelines. A detailed description of the constraints, particularly those in our experiments, is provided in Appendix D.

---

**Algorithm 1** PCFM: Physics-Constrained Flow Matching

---

**Require:** Flow model $v_\theta(u, \tau)$, constraint residual $h(u)$, initial state $u_0$, steps $N$, penalty $\lambda$
**Ensure:** Final state $u_1$ such that $h(u_1) = 0$
1: $\Delta\tau \leftarrow 1/N, \quad u \leftarrow u_0$
2: **for** $k = 0, \dots, N-1$ **do**
3: $\quad \tau \leftarrow k \cdot \Delta\tau, \quad \tau' \leftarrow \tau + \Delta\tau$
4: $\quad u_1 \leftarrow \text{ODESolve}(u,\, v_\theta,\, \tau,\, 1,\, \theta)$
5: $\quad J \leftarrow \nabla h(u_1)^\top$
6: $\quad u_{\text{proj}} \leftarrow u_1 - J^\top (JJ^\top)^{-1} h(u_1)$
7: $\quad \hat{u}_{\tau'} \leftarrow \text{ODESolve}(u_{\text{proj}},\, -(u_{\text{proj}} - u_0),\, 1,\, \tau')$
8: $\quad u_{\tau'} \leftarrow \arg\min_u \|u - \hat{u}_{\tau'}\|^2 + \lambda \|h(u + (1 - \tau')v_\theta(u_{\tau'}, \tau'))\|^2$
9: $\quad u \leftarrow u_{\tau'}$
10: **if** $\|h(u)\| > \epsilon$ **then**
11: $\quad u \leftarrow \arg\min_u \|u - u_1\|^2 \quad \text{s.t.} \quad h(u) = 0$
12: **return** $u$

---

### 3.3 PCFM: Physics-Constrained Flow Matching

Given a pretrained generative flow model $v_\theta(u, \tau)$, the flow dynamics define a pushforward map $\phi_\tau$ that transports samples $u_0 \sim \pi_0$ to solution-like outputs $u_1 = \phi_1(u_0) \sim \pi_1$. In our setting, constraints are imposed on the final sample $u_1$, via a nonlinear function $\mathcal{H}u := h(u_1) = 0$, which we wish to enforce *exactly* during inference [3]. Our goal is to generate samples $u(\tau)$ along the generative trajectory such that the terminal output $u_1$ satisfies $h(u_1) = 0$, while remaining aligned with the learned flow $v_\theta$. To achieve this, we develop a constraint-guided sampling algorithm that interleaves lightweight constraint corrections with marginally consistent flow updates. The core procedure is outlined below.

**Forward Shooting and Projection.** At each substep $\tau \to \tau + \delta\tau$, we first perform a *forward solve* (shooting) to the final flow time $\tau = 1$, $u_1 = \text{ODESolve}(u(\tau), v_\theta, \tau, 1)$, typically using an inexpensive integrator (e.g., Euler with large step size) since high precision is not required for intermediate inference. We then apply a single *Gauss–Newton projection* to softly align $u_1$ to the constraint manifold,

$$r = h(u_1), \quad J = \nabla h(u_1)^\top, \quad u_{\text{proj}} = u_1 - J^\top (JJ^\top)^{-1} r, \tag{4}$$

which shifts $u_1$ minimally to a linearized feasible point. As formalized in Proposition E.1, this projection step corresponds to a projection onto the tangent space of the constraint manifold at $u_1$, and recovers the exact solution when $h$ is affine.

---

[3] While the usage of the term *hard constraint* typically implies that the constraint residual must be exactly zero, we use it synonymously with *approximate hard constraint*, referring to constraint satisfaction up to solver and machine numerical precision, as common in related works [54, 31, 14].

**Reverse Update via OT Interpolant.** To propagate this correction back to $\tau' = \tau + \delta\tau$, we consider a reverse ODE solve $u_{\tau'} = \text{ODESolve}\big(u(\tau), -v_\theta, 1, \tau'\big)$. ODEs are inherently reversible in theory, however in practice it may have $\mathcal{O}(1)$ error in the backward integration [64] mainly because of the flip of the signs of the eigenvalues of the Jacobian of $v_\theta$ [47]. Gholami et al. showcases this can be unstable for neural ODEs with a large absolute value of the Lipschitz constant of $v_\theta$ [47, 46]. However, robust options, such as implicit or adjoint methods [48, 65], are computationally prohibitive during inference. To alleviate these issues during inference, instead, we approximate the reverse flow using the Optimal Transport (OT) [4, 11] displacement interpolant used during flow matching:

$$\hat{u}_{\tau'} = \text{ODESolve}\big(u_{\text{proj}}, -(u_{\text{proj}} - u_0), 1, \tau'\big). \tag{5}$$

This avoids instability and is justified by Proposition 3.1, which shows that the straight-line displacement approximates the true marginal flow in the small-step limit. This perspective is further supported by recent findings in Rectified Flows [36], where generative paths become increasingly linear as flow resolution improves.

**Proposition 3.1** (Reversibility under OT Displacement Interpolant)**.** *Let $v_\theta(u, \tau)$ be a pre-trained marginal velocity field learned via deterministic flow matching, with pushforward map $\phi_\tau$ such that $u_1 = \phi_1(u_0)$, for samples $u_0 \sim \pi_0$ and $u_1 \sim \pi_1$. Suppose $v_\theta(u, \tau)$ is Lipschitz continuous in both $u$ and $\tau$. Then under numerical integration with step size $\delta\tau$, the flow satisfies*

$$v_\theta(u(\tau), \tau) = u_1 - u_0 + \mathcal{O}(\delta\tau^p),$$

*where $p$ is the order of the integrator. Moreover, defining the OT displacement interpolant $\bar{v}(u) := u_1 - u_0$, the reverse update $\hat{u}_{\tau'} = \text{ODESolve}\big(u_{\text{proj}}, -\bar{v}(u), 1, \tau'\big)$ produces a numerically stable and solver-invariant approximation that is unconditionally reversible as $\delta\tau \to 0$.*

**Relaxed Constraint Correction.** Due to discretization and the nonlinearity of $h$, the point $\hat{u}_{\tau'}$ may still incur residual error. We thus perform a penalized correction:

$$u_{\tau'} = \arg\min_u \|u - \hat{u}_{\tau'}\|^2 + \lambda \big\|h(u + \gamma\, v_\theta(u, \tau'))\big\|^2, \quad \gamma = 1 - \tau', \tag{6}$$

which encourages constraint satisfaction at the extrapolated point while preserving local flow alignment. As $\delta\tau \to 0$, this step aligns with the OT-based marginal objective, and offers robustness in settings with coarse discretization, for e.g., necessitated by the need of lower number of function evaluations. We provide an ablation in Appendix L.3 highlighting its effect.

**Final Projection and Runtime.** We set $u_{\tau+\delta\tau} := u_{\tau'}$, and iterate. If constraint residuals remain high at $\tau = 1$, we apply a final full projection:

$$u_1 = \arg\min_{u'} \|u' - u_1\|^2 \quad \text{subject to} \quad h(u') = 0. \tag{7}$$

To this end, we have developed a custom, batched differentiable solver that performs these updates entirely in parallel: at each iteration it assembles and inverts only an $m \times m$ Schur complement system (with $m = \dim h \ll n$), then applies a Newton-style correction to the full state (see Appendix I for the specific algorithm details). Despite the extra iterations, the overall cost remains $\mathcal{O}(n)$ per sample, since (i) each Schur solve is $\mathcal{O}(m^3)$ with $m \ll n$, (ii) the backward flow solve in Eq. (5) is a single Euler step, and (iii) the relaxed-penalty correction in Eq. (6) typically converges in 3–5 gradient steps. We also provide a detailed runtime analysis in Appendix E.1, showing that the final projection step contributes only 1–3% of total time; most cost is due to sampling itself.

We note that ECI [12] emerges as a special case of this framework when constraints are linear and non-overlapping, and the penalty is omitted ($\lambda = 0$) (see Appendix F). Our method generalizes this by enabling strict enforcement of nonlinear and overlapping constraints while preserving generative consistency. A separate ablation study is also provided highlighting the necessity and role of constraint-guided sampling with consistent flow updates in Appendix L.1.

### 3.4 Reversibility and Interpolant Compatibility

Our method naturally extends to other interpolants, including variance-preserving (VP) Score-Based Diffusion Models (SBDMs) [2, 4]. While OT interpolants yield straight-line paths, VP flows benefit from bounded Lipschitz constants due to Gaussian smoothing, making them more stable in reverse [66]. The key requirement is access to a consistent displacement direction (e.g., $u_1 - u_0$), which enables constraint enforcement via reversible approximations regardless of the underlying interpolant.

Table 3: Generative performance for zero-shot methods on constrained PDEs with linear and nonlinear constraints. Heat and Navier-Stokes enforce global conservation laws (CL) as linear constraints, along with initial condition (IC) constraints. In contrast, Burgers and Reaction-Diffusion apply CL as nonlinear constraints along with IC or BC constraints. Lower values indicate better performance, and **best results are highlighted in bold.**

| Dataset | Metric | PCFM | ECI | DiffusionPDE | D-Flow | PDM | FFM |
|---|---|---|---|---|---|---|---|
| Heat Equation | MMSE / $10^{-2}$ | **0.241** | 0.697 | 4.49 | 1.97 | 0.45 | 4.56 |
| | SMSE / $10^{-2}$ | 0.937 | 0.973 | 3.93 | 1.14 | **0.02** | 3.51 |
| | CE (IC) / $10^{-2}$ | **0** | **0** | 599 | 102 | **0** | 579 |
| | CE (CL) / $10^{-2}$ | **0** | **0** | 2.06 | 64.8 | **0** | 2.11 |
| | FPD | 1.22 | 1.34 | 1.70 | 2.70 | **0.17** | 1.77 |
| Navier-Stokes | MMSE / $10^{-2}$ | **4.59** | 5.23 | 17.4 | – | 12.21 | 16.5 |
| | SMSE / $10^{-2}$ | **4.17** | 7.28 | 9.48 | – | 6.61 | 7.90 |
| | CE (IC) / $10^{-2}$ | **0** | **0** | 288 | – | **0** | 328 |
| | CE (CL) / $10^{-2}$ | **0** | **0** | 21.4 | – | **0** | 18.6 |
| | FPD | **1.00** | 1.04 | 3.70 | – | 2.01 | 2.81 |
| Reaction-Diffusion IC | MMSE / $10^{-2}$ | **0.026** | 0.324 | 3.16 | 0.318 | 1.74 | 2.92 |
| | SMSE / $10^{-2}$ | 0.583 | **0.060** | 2.54 | 6.86 | 0.43 | 2.54 |
| | CE (IC) / $10^{-2}$ | **0** | **0** | 451 | 215 | **0** | 445 |
| | CE (CL) / $10^{-2}$ | **0** | 6.00 | 3.82 | 29.7 | **0** | 3.87 |
| | FPD[†] | **15.7** | 136 | 44.1 | 28.3 | 109 | 24.9 |
| Burgers BC | MMSE / $10^{-2}$ | 0.335 | 0.359 | 5.42 | **0.224** | 11.8 | 4.86 |
| | SMSE / $10^{-2}$ | 0.123 | **0.089** | 1.30 | 0.948 | 2.67 | 1.38 |
| | CE (BC) / $10^{-2}$ | **0** | 20.3 | 426 | 95.7 | **0** | 409 |
| | CE (CL) / $10^{-2}$ | **0** | 15.7 | 6.20 | 15.0 | **0** | 6.91 |
| | FPD | **0.292** | 0.307 | 25.9 | 1.44 | 6.41 | 24.7 |
| Burgers IC | MMSE / $10^{-2}$ | **0.052** | 10.0 | 14.3 | 9.97 | 153 | 13.7 |
| | SMSE / $10^{-2}$ | **0.272** | 6.65 | 8.06 | 7.91 | 2.62 | 7.90 |
| | CE (IC) / $10^{-2}$ | **0** | **0** | 471 | 397 | **0** | 462 |
| | CE (CL) / $10^{-2}$ | **0** | 205 | 6.22 | 8.66 | **0** | 6.91 |
| | FPD | **0.101** | 1.31 | 35.8 | 22.1 | 99.7 | 33.5 |

[†] *Pretrained Poseidon requires spatially square inputs; we bilinearly interpolated solution grids to 128×128, which may introduce artifacts in FPD evaluation. D-Flow and PDM results are omitted due to numerical instabilities.*

[†] *Pretrained Poseidon requires spatially square inputs; we bilinearly interpolated solution grids to 128×128, which may introduce artifacts in FPD evaluation. D-Flow results are omitted due to numerical instabilities.*

### 3.4.1 Tradeoff Between Flow Steps and Relaxed Constraint Correction

In PCFM, the number of flow steps and the strength of the relaxed constraint correction jointly determine the quality of the final solution. Using fewer steps speeds up inference but increases numerical error, potentially violating constraints at intermediate states. In such regimes, the relaxed correction term in Equation (6) provides a mechanism to compensate for deviation from the constraint manifold. Conversely, with sufficiently many flow steps, the penalty term becomes less critical, and the dynamics naturally stay closer to the OT path [36]. We present detailed ablation results in Appendix L.3, showing how PCFM adapts across this tradeoff.

## 4 Experiments and Results

In this section, we highlight the key results and compare our approach with representative baselines. For every problem, we construct a PDE numerical solution dataset with two degrees of freedom by varying initial (IC) and boundary conditions (BC) (see Appendix B), and pre-train an uncondi-tional FFM model [11] on this dataset (see Appendix C). To evaluate generative performance and constraint satisfaction, we focus on generating solution subsets constrained on a selected held-out IC or BC, aiming to guide the pretrained model toward the corresponding solution subset. Fur-thermore, we incorporate physical constraints, specifically global mass conservation, via PCFM and adapt other baseline methods, where possible, through their sampling frameworks to evaluate

their performance on constraint satisfaction tasks. We open-source our code: a Python implementation at `https://github.com/cpfpengfei/PCFM` and an experimental Julia implementation at `https://github.com/utkarsh530/PCFM.jl`.

For a variety of PDEs with different constraint requirements, ranging from easier linear constraints to harder nonlinear constraints, we evaluate different sampling methods using the same pretrained FFM model. The specific linear and nonlinear constraints used in our PCFM approach are outlined in Appendix D, while the parameters and set-up for other methods are outlined in Appendix H. To evaluate the similarity between the ground truth and generated solution distributions, we follow Kerrigan et al. [11] and Cheng et al. [12] to compute the pointwise mean squared errors of the mean (**MMSE**) and standard deviation (**SMSE**), along with the Fréchet Poseidon Distance (**FPD**). The FPD quantifies distributional similarity in feature space using a pretrained PDE foundation model (Poseidon) [67], analogous to Fréchet Inception Distance (FID) used in image generation [68]. Importantly, we evaluate the **constraint errors** by taking the $\ell_2$ norm of the residuals for initial condition (IC), boundary condition (BC), and global mass conservation (CL) over time $t$, averaged across all $N$ generated samples: $\text{CE}(*) = \frac{1}{N} \sum_{n=1}^{N} \left\| \mathcal{R}_* \left( \hat{u}^{(n)} \right) \right\|_2$, where $* \in \{\text{IC}, \text{BC}, \text{CL}\}$

## 4.1 Linear Hard Constraints

We first focus on linear hard constraints by considering the 2-D Heat equation ($u(x, t)$) and the 3-D Navier-Stokes ($u(x, y, t)$) equation with periodic boundary conditions. The global conservation laws simplify to a linear integral over the solution field, where the conserved mass quantity reduces to a sum or mean of the solution across the spatial domain (see Appendix D). We aim to constrain the generation to satisfy a selected IC and global mass conservation. In Table 3, PCFM outperforms all the compared methods in the MMSE, SMSE, and FPD, achieving machine-level precision for constraints on both IC and global mass conservation. Due to the simplicity of the linear constraints, ECI is also capable of achieving constraint satisfaction in generated samples.

PCFM further enables us to improve the performance by applying relaxed constraint corrections (Equation 6), achieving higher solution fidelity with fewer flow matching steps. For high-dimensional settings such as Navier–Stokes, we adopt a stochastic interpolant perspective by randomizing $u_0$ across batches [37]. This enhances alignment with the OT displacement path while preserving approximate hard constraint satisfaction at generation time.

## 4.2 Nonlinear Hard Constraints

### 4.2.1 Nonlinear Global and Boundary Constraints

**Reaction-Diffusion (nonlinear mass & Neumann flux).** We first consider a reaction-diffusion PDE with nonlinear mass conservation and Neumann boundary fluxes. When constrained on the fixed IC and nonlinear mass, PCFM achieves the lowest constraint errors and best generation fidelity (Table 3, Figure 2). ECI meets the IC constraint by exact value enforcement in their correction step but its framework cannot enforce nonlinear mass conservation. DiffusionPDE and D-Flow, despite using a combined loss function on IC and PINN-loss, likewise fail to enforce both hard constraints simultaneously. In contrast, PCFM satisfies IC and mass conservation to machine precision for all $t$ (Figure 2), which in turn enforces Neumann boundary fluxes and improves overall solution quality.

**Burger's Equation (nonlinear mass & Dirichlet BCs).** Constraining Dirichlet BC at $x = 0$ and zero-flux Neumann BC at $x = 1$, generated samples should satisfy both BC and nonlinear mass conservation constraints for all $t$. PCFM achieves constraint satisfaction while matching similar fidelity of other methods (Figure 5).

### 4.2.2 Nonlinear Local Dynamics and Shock Constraints

We demonstrate PCFM's ability to tackle a more challenging task of enforcing global nonlinear and local dynamical constraints, achieving improved generation quality while enforcing physical consistencies. Specifically, for Burgers' equation constrained on ICs, PCFM outperforms all baselines across metrics while satisfying both IC and mass conservation. In addition to global constraints, we incorporate 5 unrolled local flux collocation points in the residual (see Appendix D for further details). Projecting intermediate $u_1$ to the constraint manifold helps capture the localized shock structure (see top left corner of Figure 3), improving solution fidelity to match the shock dynamics in Burgers. In contrast, other methods cannot satisfy hard constraints nor capture the shock dynamics.

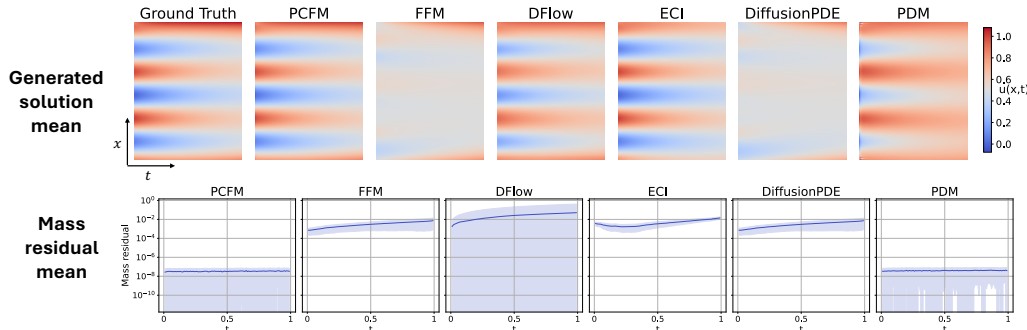

Figure 2: Comparison of mean $\pm$ 1 std. of mass residuals across samples. generated solutions and mass conservation errors for the Reaction-Diffusion problem with IC fixed. By enforcing both IC and nonlinear mass conservation constraints, PCFM improves quality of generated solutions while satisfying both constraints exactly.

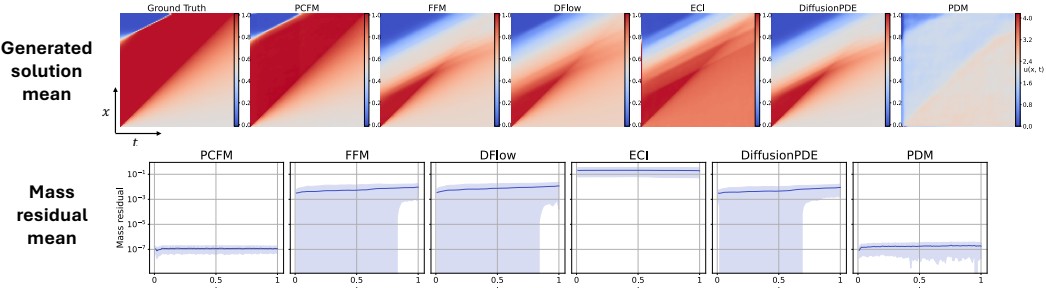

Figure 3: Comparison of mean generated solutions and mass conservation errors for the Burger's problem with IC fixed. By enforcing nonlinear conservation constraints via PCFM, our method captures the Burgers' shock phenomenon, ensures global mass conservation in the generated solution, while improving solution quality. Shaded bands show $\pm$ 1 std. of mass residuals across samples.

### 4.3 Enhancing Fidelity through Additional Physical Constraints

In our ablation study (Figure 4), we investigate the effect of imposing more constraints, specifically the effect of constrained collocation points, in addition to the IC and mass conservation constraints in the Burgers' problem. We show that adding more constrained collocation points improves generation quality without worsening the satisfaction of IC and mass constraints (see Appendix D for constraint set details). Interestingly, while stacking multiple constraints increases computational cost in each Gauss-Newton projection, we find that complementary constraints (local flux, IC, and global mass) can improve performance, unlike soft-constraint approaches such as PINNs, where adding more constraints can degrade performance due to competing objectives [26, 25]. However, in cases of conflicting constraints, tradeoffs will arise that require careful balance of their effects on generation fidelity and physical consistency. Indeed, the ability to chain different hard constraints on a pretrained generative model makes PCFM flexible and practical for diverse applications.

We also explored total variation diminishing (TVD) constraints to promote smoother solution profiles, demonstrating on the heat equation that TVD improves both smoothness and fidelity while preserving IC and global mass constraints (see Appendix L). This highlights PCFM's flexibility to incorporate multiple variety of constraints to enhance generative quality.

## 5 Conclusion

We presented PCFM, a zero-shot inference framework for enforcing arbitrary nonlinear equality constraints in pretrained flow-based generative models. PCFM combines flow-based integration, tangent-space projection, and relaxed penalty corrections to strictly enforce constraints while staying consistent with the learned generative trajectory. Our method supports both local and global constraints, including conservation laws and consistency constraints, without requiring retraining or architectural changes. Empirically, PCFM outperforms state-of-the-art baselines across diverse PDEs,

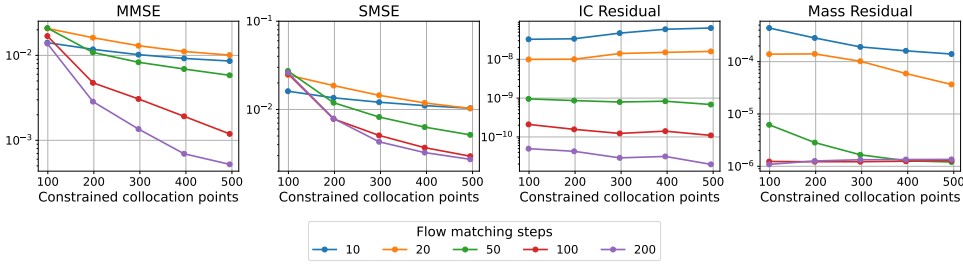

Figure 4: Increasing the number of constraints (constraint collocation points) can improve solution fidelity while maintaining strong satisfaction of other constraints (IC and global mass conservation), demonstrating the ability of PCFM to handle chaining of multiple constraints.

including systems with shocks and nonlinear dynamics, achieving lower error and exact constraint satisfaction. Notably, we find that enforcing additional complementary constraints improves generation quality, contrary to common limitations observed in soft-penalty methods such as PINNs [25, 26]. While `PCFM` currently focuses on equality constraints, extending it to inequality constraints and leveraging structure in constraint Jacobians are promising directions for improving scalability [69, 50]. Our work offers a principled approach for strictly enforcing physical feasibility in generative models, with broad impact on scientific simulation and design.

## 6    Acknowledgments

We acknowledge the MIT SuperCloud and Lincoln Laboratory Supercomputing Center for providing HPC resources. We also acknowledge the Delta GPU system at the National Center for Supercomputing Applications (NCSA), supported by the National Science Foundation and the ACCESS program. P.C. is supported by the Regenerative Energy-Efficient Manufacturing of Thermoset Polymeric Materials (REMAT) Energy Frontier Research Center, funded by the U.S. Department of Energy, Office of Science, Basic Energy Sciences under award DE-SC0023457.

This material is based upon work supported by the U.S. National Science Foundation under award Nos CNS-2346520, PHY-2028125, RISE-2425761, DMS-2325184, OAC-2103804, and OSI-2029670, by the Defense Advanced Research Projects Agency (DARPA) under Agreement No. HR00112490488, by the Department of Energy, National Nuclear Security Administration under Award Number DE-NA0003965 and by the United States Air Force Research Laboratory under Cooperative Agreement Number FA8750-19-2-1000. Neither the United States Government nor any agency thereof, nor any of their employees, makes any warranty, express or implied, or assumes any legal liability or responsibility for the accuracy, completeness, or usefulness of any information, apparatus, product, or process disclosed, or represents that its use would not infringe privately owned rights. Reference herein to any specific commercial product, process, or service by trade name, trademark, manufacturer, or otherwise does not necessarily constitute or imply its endorsement, recommendation, or favoring by the United States Government or any agency thereof. The views and opinions of authors expressed herein do not necessarily state or reflect those of the United States Government or any agency thereof." The views and conclusions contained in this document are those of the authors and should not be interpreted as representing the official policies, either expressed or implied, of the United States Air Force or the U.S. Government.

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

# A  Proof of Constraint Satisfaction

We now show that, under standard regularity assumptions, PCFM (see Algorithm 1) returns a final sample $u_1$ satisfying the hard constraint $h(u_1) = 0$ to numerical precision.

**Theorem A.1** (Exact Constraint Enforcement). *Let $h : \mathbb{R}^n \to \mathbb{R}^m$ be a twice continuously differentiable constraint function, and suppose the Jacobian $J_h(u) := \nabla h(u) \in \mathbb{R}^{m \times n}$ has full row rank $m \leq n$ in a neighborhood of the constraint manifold $\mathcal{M} := \{u \in \mathbb{R}^n : h(u) = 0\}$. Then the final sample $u_1$ produced by PCFM satisfies*

$$h(u_1) = 0$$

*to machine precision, provided the final projection step in Algorithm 1 is solved using sufficiently many Newton–Schur iterations.*

*Proof.* We consider the final correction step in Algorithm 1:

$$u_1 := \arg\min_{u'} \|u' - u_1\|^2 \quad \text{s.t.} \quad h(u') = 0,$$

which seeks to project the current sample $u_1$ (obtained after flow integration and relaxed penalty correction) onto the feasible manifold.

Let $u^{(0)} := u_1$ denote the initial guess to the Newton solver. The constrained root-finding problem $h(u) = 0$ is nonlinear, but under the full-rank Jacobian assumption, the *implicit function theorem* ensures a locally unique root $u^* \in \mathcal{M}$ near $u^{(0)}$. We solve the nonlinear system using the following Newton–Schur iteration:

$$u^{(k+1)} = u_1 - J_h(u^{(k)})^\top \left[ J_h(u^{(k)}) J_h(u^{(k)})^\top \right]^{-1} \left( h(u^{(k)}) + J_h(u^{(k)})(u_1 - u^{(k)}) \right). \quad \text{(A.1)}$$

This corresponds to the batched Schur-complement Newton update implemented in our differentiable solver. Standard convergence theory (e.g., Theorem 10.1 in [70]) guarantees that, for sufficiently small $\|h(u^{(0)})\|$, the iterates $u^{(k)}$ converge *quadratically* to $u^*$. Hence, for any tolerance $\varepsilon > 0$, there exists a finite $K$ such that

$$\|h(u^{(K)})\| < \varepsilon.$$

Since we continue iterations until $\|h(u^{(K)})\| < \varepsilon_{\text{tol}}$ (with $\varepsilon_{\text{tol}}$ typically set to machine precision), we conclude

$$h(u_1) = h(u^{(K)}) \approx 0.$$

Finally, we verify that the initial guess $u^{(0)} = u_1$ is within the *basin of convergence*. Each prior step of PCFM includes: (i) a Gauss–Newton projection, (ii) an OT-based backward solve, and (iii) a relaxed penalty correction. These ensure the incoming residual $\|h(u_1)\|$ is already small (typically $< 10^{-3}$). Combined with the well-conditioned Jacobian $J_h$, we ensure convergence of the final Newton–Schur projection.

$\square$

# B  PDE Dataset Details

We evaluate our method on four representative PDE systems commonly studied in scientific machine learning: the **Heat Equation**, **Navier-Stokes Equation**, **Reaction-Diffusion Equation**, and **Burgers' Equation**. These problems cover linear and nonlinear dynamics, have smooth or discontinuous solutions, and include a variety of initial and boundary condition types.

## B.1  Heat Equation

We consider the one-dimensional heat equation with periodic boundary conditions, following the setting in Hansen et al. [14]:

$$u_t = \alpha u_{xx}, \quad x \in [0, 2\pi], \quad t \in [0, 1], \quad (8)$$

with the sinusoidal initial condition

$$u(x, 0) = \sin(x + \varphi), \quad (9)$$

and periodic boundary conditions $u(0,t) = u(2\pi, t)$. The analytical solution is given by $u(x,t) = \exp(-\alpha t)\sin(x + \varphi)$. To pretrain our FFM model, we construct the dataset by varying the diffusion coefficient $\alpha \sim \mathcal{U}(1,5)$ and phase $\varphi \sim \mathcal{U}(0, \pi)$. All solutions are on a $100 \times 100$ spatiotemporal grid. During constrained sampling, we restrict to the same initial condition by fixing $\varphi = \pi/4$. The global mass conservation constraint is enforced via

$$\int_0^{2\pi} u(x,t)\,dx = 0, \quad \forall t \in [0,1]. \tag{10}$$

## B.2  Navier-Stokes Equation

We consider the 2D incompressible Navier–Stokes (NS) equation in vorticity form with periodic boundary conditions, following Li et al. [23]:

$$\partial_t w(x,t) + u(x,t) \cdot \nabla w(x,t) = \nu \Delta w(x,t) + f(x), \tag{11}$$
$$\nabla \cdot u(x,t) = 0, \tag{12}$$

where $w = \nabla \times u$ denotes the vorticity and $\nu = 10^{-3}$ is the viscosity. The initial vorticity $w_0$ is sampled from a Gaussian random field, and the forcing function is defined as: $f(x) = 0.1\sqrt{2}\sin\left(2\pi(x_1 + x_2) + \phi\right)$, $\phi \sim \mathcal{U}(0, \pi/2)$. We solve the NS system using a Crank–Nicolson spectral solver on a $64 \times 64$ periodic grid, recording 50 uniformly spaced temporal snapshots over the interval $T = 49$. For training, we generate 10000 simulations by sampling 100 random initial vorticities and 100 forcing phases. For test dataset, we sample an additional 10 vorticities and 100 forces, yielding 1000 solutions. We fix to 1 initial vorticity for comparison during sampling.

Here, we prove the global mass conservation for this NS problem set-up. Under periodic boundary conditions, global vorticity is conserved. Specifically, integrating the vorticity equation over the spatial domain $\Omega$ and applying the divergence theorem yields:

$$\frac{d}{dt}\int_\Omega w(x,t)\,dx = \int_\Omega \nu\Delta w(x,t) + f(x) - u \cdot \nabla w(x,t)\,dx = 0,$$

since the divergence and Laplacian terms vanish due to periodicity and the incompressibility condition $\nabla \cdot u = 0$. Thus, the total vorticity $\int_\Omega w(x,t)\,dx$ remains constant for all $t$.

## B.3  Reaction-Diffusion Equation

We consider a nonlinear 1D reaction-diffusion equation with Neumann boundary conditions:

$$u_t = \rho u(1 - u) - \nu \partial_x u, \tag{13}$$

on the domain $x \in [0,1], t \in [0,1]$, where $(\rho, \nu) = (0.01, 0.005)$. The boundary fluxes at the left and right ends are specified as $g_L$ and $g_R$, respectively. The initial condition $u(x,0)$ is sampled from a randomized combination of sinusoidal and localized bump functions (see Appendix D for full specification). The training dataset is constructed by pairing 80 initial conditions with 80 boundary conditions, producing 6400 PDE solutions, all discretized on a $(nx, nt) = (128, 100)$ space-time grid. We use a semi-implicit finite difference solver with CFL-controlled time stepping. During constrained sampling, we fix one initial condition and vary the boundary fluxes to enforce constraint satisfaction.

As this system involves nonlinear source and flux terms, global conservation is no longer trivial. Integrating both sides of the PDE over the domain yields:

$$\frac{d}{dt}\int_0^1 u(x,t)\,dx = \rho\int_0^1 u(1-u)\,dx + g_L(t) - g_R(t), \tag{14}$$

which depends nonlinearly on the state $u(x,t)$ and boundary fluxes $g_L, g_R$. Thus, conservation must be tracked through both the nonlinear reaction term and boundary-driven transport, making hard constraint enforcement via our framework necessary.

## B.4  Inviscid Burgers' Equation

We consider the 1D inviscid Burgers' equation with mixed Dirichlet and Neumann boundary conditions:

$$u_t + \frac{1}{2}(u^2)_x = 0, \quad x \in [0,1], \quad t \in [0,1], \tag{15}$$

with initial condition defined as a smoothed step function centered at a random location $p_{\mathrm{loc}} \sim \mathcal{U}(0.2, 0.8)$:

$$u(x, 0) = \frac{1}{1 + \exp\left(\frac{x - p_{\mathrm{loc}}}{\epsilon}\right)}, \quad \epsilon = 0.02. \tag{16}$$

We apply a Dirichlet condition on the left, $u(0, t) = u_{\mathrm{bc}}$ with $u_{\mathrm{bc}} \sim \mathcal{U}(0, 1)$, and a Neumann condition (zero-gradient) on the right. Solutions are computed using a Godunov finite volume method on a $(nx, nt) = (101, 101)$ grid. For pretraining the FFM model, we vary both initial and boundary conditions, generating 80 variants each to create 6400 solutions in the training set. For constrained sampling, we use two configurations: (1) fixing the initial condition and varying the boundary condition, and (2) fixing the boundary condition and varying the initial condition.

The conservation constraint is nonlinear and particularly sensitive due to shock formation. Integrating the PDE yields

$$\frac{d}{dt} \int_0^1 u(x, t)\, dx = -\left[\tfrac{1}{2} u(1, t)^2 - \tfrac{1}{2} u(0, t)^2\right],$$

so global conservation requires $u(1, t)^2 = u(0, t)^2$, making the constraint highly sensitive to boundary-induced discontinuities.

## C  FFM Model Pretraining

For all PDE benchmarks, we employ the functional flow matching (FFM) [11] framework and parameterize the underlying time-dependent vector field with a Fourier neural operator (FNO) backbone [23]. The FNO encoder takes the input the concatenation of the current state $u_\tau$, a sinusoidal positional embedding for the spatial grid and Fourier time embedding. For each benchmark, we adopt the following training scheme unless otherwise specified:

- Optimizer: Adam optimizer with learning rate $3 \times 10^{-4}$, $\beta_1 = 0.9$, $\beta_2 = 0.999$, and no weight decay.

- Learning rate scheduler: Reduce-on-plateau scheduler with a factor of $0.5$, patience of $10$ validation steps, and a minimum learning rate of $1 \times 10^{-4}$.

- Batch size: 256 for 1D problems (Heat, Reaction-Diffusion, Burgers) and 24 for the 2D Navier-Stokes problem.

For Heat, we follow Cheng et al. [12] to train on 5000 analytical solutions of the heat equation. For Reaction-Diffusion and Burgers, we train each model on 6400 numerical PDE solutions. For these 3 problems, the FNO is configured with 4 Fourier layers with 32 Fourier modes for both spatial and temporal dimensions, 64 hidden channels, and 256 projection channels. For Navier-Stokes equation, we train a 3D FFM on $10,000$ numerical solutions. The FNO has 2 Fourier layers with 16 Fourier modes per dimension, 32 hidden channels, and 256 projection channels, following Cheng et al. [12]. FFM models for Heat, Reaction-Diffusion, and Burgers are each trained over $20,000$ steps on 1 NVIDIA V100 GPU and the Navier-Stokes model is trained for $500,000$ steps on 4 NVIDIA A100 GPUs.

Following Kerrigan et al. [11], all FFM models use a Gaussian prior noise $\mathcal{P}(u_0)$. For all 1D problems, we adopt a *Matern Gaussian Process* prior with smoothness parameter $\nu = 0.5$, kernel length $0.001$, and variance $1.0$, implemented using `GPyTorch`. For Navier-Stokes (2D), we adopt standard Gaussian white noise as the prior, $\mathcal{P}(u_0) = \mathcal{N}(0, I)$. Importantly, PCFM is applied post-hoc at inference time, allowing us to reuse the pretrained unconditional FFM models across constraint configurations.

## D  Constraints Enforcement via PCFM

We focus on physical and geometric constraints of the form $\mathcal{H}u = 0$, which must be satisfied as equality conditions on the solution field $u$. First, we describe common equality constraints encountered in solving PDEs. Next, we describe specific constraint setup for our PDE datasets.

### D.1 Linear Equality Constraints

**Dirichlet Initial and Boundary Conditions**   Constraints such as $u(x, 0) = \alpha_0(x)$ or $u(x, t) = g(x, t)$ on $\partial\Omega \times [0, T]$ define fixed-value conditions that are linear in $u$. These can be encoded in the general form

$$\mathcal{H}u = Au - b = 0, \tag{17}$$

where $A$ is a sampling or interpolation operator applied to $u$, and $b$ is the prescribed target value.

**Linear Global Conservation Laws (Periodic Systems)**   In systems with periodic boundary conditions, additive invariants such as total mass are often linearly conserved. A representative example is

$$\mathcal{H}u = \int_\Omega u(x, t)\, dx - C = 0, \tag{18}$$

where $C \in \mathbb{R}$ is the conserved quantity. Such constraints can be exactly enforced via analytically derived projections [14], geometric integrators [61], and numerical optimization [69, 50].

### D.2 Nonlinear Equality Constraints

Many physical systems exhibit global conservation laws where the conserved quantity depends nonlinearly on $u$. A representative form is:

$$\mathcal{H}u = \frac{d}{dt} \int_\Omega \rho(u(x, t))\, dx = 0, \tag{19}$$

where $\rho(u)$ denotes a nonlinear density, such as total energy or entropy. While such constraints are often not conserved exactly by standard numerical integrators, structure-preserving methods, such as symplectic or variational integrators [61], can conserve invariants approximately over long time horizons. In our setting, however, we seek to enforce such nonlinear conservation laws exactly at inference time and moreover only at the final denoised sample. Thus, we rely on applying iterative projection methods such as Newton's method for constrained least squares [69] to satisfy Equation (19) up to numerical tolerance.

**Neumann-Type Boundary Constraints**   Boundary constraints involving fluxes, such as

$$\mathcal{H}u = \partial_n u(x, t) - g(x, t) = 0, \tag{20}$$

are typically linear under standard semi-discretizations, as the derivative operator is linear. However, these constraints can become effectively nonlinear in practice when implemented with upwinding or other nonlinear flux-discretization schemes, particularly in hyperbolic PDEs [62]. Additionally, if the prescribed flux $g$ depends nonlinearly on $u$, the constraint becomes explicitly nonlinear. In both cases, we handle them using differentiable correction procedures or projection-based updates within our framework.

In our PCFM framework, the residual operator $\mathcal{H}(u)$ specifies the equality constraints to be enforced on the generated solution $u$. We enforce these constraints on the generated solution state in our PCFM projection operator, ensuring that the final solution strictly satisfies the required physical constraints without retraining the underlying flow model. We summarize the specific constraint formulations used for each PDE task below.

### D.3 Heat Equation (IC and Mass Conservation)

The residual is a concatenation of two linear constraints:

$$\mathcal{H}_{\text{Heat}}(u) = \begin{bmatrix} u(x, t = 0) - u_{\text{IC}}(x) \\ \int u(x, t)\, dx - \int u(x, t = 0)\, dx \end{bmatrix} \tag{21}$$

where the first term enforces the initial condition and the second term ensures mass conservation over time.

### D.4 Navier-Stokes Equation (IC and Mass Conservation)

For the 2D Navier-Stokes problem, we impose the initial vorticity and the global mass conservation constraint over the spatial domain:

$$\mathcal{H}_{\text{NS}}(u) = \left[ \begin{array}{c} u(x,y,t=0) - u_{\text{IC}}(x,y) \\ \iint u(x,y,t)\,dx\,dy - \iint u(x,y,t=0)\,dx\,dy \end{array} \right] \tag{22}$$

where $u(x,y,t)$ represents the vorticity field over spatial coordinates $(x,y)$ and time $t$. The mass conservation term enforces that the total vorticity integrated over the spatial domain remains consistent with the initial condition over all time steps.

### D.5 Reaction-Diffusion (IC and Nonlinear Mass Conservation)

We enforce both the initial condition and nonlinear mass conservation which accounts for reaction and boundary flux terms:

$$\mathcal{H}_{\text{RD}}(u) = \left[ \begin{array}{c} u(x,t=0) - u_{\text{IC}}(x) \\ m(t) - \left( m(0) + \int_0^t \rho(u)\,dt + \int_0^t (g_L - g_R)\,dt \right) \end{array} \right] \tag{23}$$

where $m(t) = \int u(x,t)\,dx$ is the total mass, $\rho(u) = \rho u(1-u)$ is the nonlinear reaction source term, and $g_L, g_R$ are the Neumann boundary fluxes.

### D.6 Burgers (BC and Mass Conservation)

We enforce both the boundary conditions (Dirichlet and Neumann) and nonlinear mass conservation:

$$\mathcal{H}_{\text{Burgers-BC}}(u) = \left[ \begin{array}{c} u(x=0,t) - u_L \\ u(x=-1,t) - u(x=-2,t) \\ \int u(x,t)\,dx - \int u(x,t=0)\,dx \end{array} \right] \tag{24}$$

where $u_L$ is the Dirichlet boundary value at the left boundary, and the Neumann BC enforces a zero-gradient at the right boundary.

### D.7 Burgers (IC, Mass Conservation, and Local Flux)

We impose three complementary constraints for the Burgers equation: (i) the initial condition, (ii) global nonlinear mass conservation, and (iii) a sequence of local conservation updates based on Godunov's flux method. Specifically, the residual is formulated as:

$$\mathcal{H}_{\text{Burgers-IC}}(u) = \left[ \begin{array}{c} u(x,t=0) - u_{\text{IC}}(x) \\ \mathcal{R}_{\text{Flux}}^{(k)}(u) \\ \int u(x,t)\,dx - \int u(x,t=0)\,dx \end{array} \right] \tag{25}$$

where $\mathcal{R}_{\text{Flux}}^{(k)}(u)$ applies $k$ unrolled discrete updates (collocation points) based on Godunov flux:

$$\mathcal{F}(u_L, u_R) = \begin{cases} \min\left( \frac{1}{2}u_L^2, \frac{1}{2}u_R^2 \right) & \text{if } u_L \leq u_R \\ \frac{1}{2}u_L^2 & \text{if } u_L > u_R \text{ and } \frac{u_L + u_R}{2} > 0 \\ \frac{1}{2}u_R^2 & \text{if } u_L > u_R \text{ and } \frac{u_L + u_R}{2} \leq 0 \end{cases} \tag{26}$$

We apply $k \in \{1, \ldots, 5\}$ unrolled steps to incrementally enforce local conservation dynamics alongside global mass conservation and initial condition satisfaction.

## E   Details of the PCFM

### E.1 Runtime Analysis and Overhead

Assessing runtime and memory overhead is essential for evaluating the practical feasibility of PCFM. We therefore report detailed runtime and memory comparisons in Table 4 and Table 5 across all

baselines using consistent batch sizes (8 for Navier–Stokes and 32 for other datasets) and 100 flow-matching steps on a single NVIDIA V100 GPU. To ensure a fair and interpretable comparison, all experiments are conducted using $\lambda = 0$ in Equation (6). The choice of $\lambda$ can significantly affect convergence behavior and computational cost, as larger penalty weights introduce additional correction steps (a separate ablation is available in Appendix L.3). However, we observe that $\lambda = 0$ is sufficient to achieve state-of-the-art performance in most cases, while also isolating the intrinsic cost of our projection method with greater clarity.

PCFM introduces a modest overhead relative to unconstrained baselines such as vanilla FFM, owing to the additional projection step for enforcing constraints. Nevertheless, it achieves a 4–6× speedup over D-Flow [35], which requires costly backpropagation through the entire ODE unroll. DiffusionPDE [10] runs faster but fails to enforce nonlinear constraints robustly, particularly for problems such as Burgers and Reaction–Diffusion. ECI exhibits runtime growth with the number of mixing steps but similarly struggles with nonlinear or coupled constraints. For linear systems such as Heat and Navier–Stokes, the projection converges rapidly, resulting in comparable runtimes to PDM [49]. Overall, PCFM incurs a small yet necessary computational overhead to achieve robust constraint enforcement, maintaining competitive runtime efficiency while providing strict constraint satisfaction.

Table 4: Runtime and memory comparison across methods. Reported values correspond to per-sample inference on a single NVIDIA V100 32 GB GPU.

| Dataset | Metric | PCFM | ECI | DiffusionPDE | D-Flow | PDM | Vanilla |
|---|---|---|---|---|---|---|---|
| Heat | Time (ms/sample) | 291.8 | 65.6 | 128.6 | 2770.5 | 361.3 | 65.2 |
| | Peak Memory (GB) | 2.41 | 1.21 | 2.39 | 3.58 | 1.06 | 1.21 |
| Burgers | Time (ms/sample) | 1371.0 | 780.6 | 211.2 | 8048.9 | 444.9 | 105.4 |
| | Peak Memory (GB) | 2.57 | 1.26 | 2.43 | 3.65 | 1.08 | 1.22 |
| Reaction–Diffusion | Time (ms/sample) | 636.6 | 744.4 | 159.9 | 6578.7 | 474.7 | 80.7 |
| | Peak Memory (GB) | 2.97 | 1.72 | 2.93 | 4.33 | 1.43 | 1.43 |
| Navier–Stokes | Time (ms/sample) | 350.7 | 3355.6 | 678.8 | 16447.2 | 338.7 | 343.8 |
| | Peak Memory (GB) | 7.39 | 2.03 | 3.90 | 5.85 | 1.93 | 2.02 |

Table 5: Runtime breakdown for PCFM showing the cost of sampling and the final projection step.

| Dataset | Component | Time (ms/sample) | Percentage |
|---|---|---|---|
| Heat Equation | Sampling | 286.6 | 98.22% |
| | Final Projection | 5.19 | 1.78% |
| Burgers Equation | Sampling | 1358.8 | 99.11% |
| | Final Projection | 12.2 | 0.89% |
| Reaction–Diffusion | Sampling | 630.2 | 98.99% |
| | Final Projection | 6.40 | 1.01% |
| Navier–Stokes | Sampling | 350.0 | 99.8% |
| | Final Projection | 0.7 | 0.2% |

## E.2 Tangent-Space Interpretation of the Projection Step

We provide additional theoretical context for the PCFM. In Proposition E.1, we formalize the projection step as a tangent-space update in Hilbert spaces and show that it corresponds to an orthogonal projection onto the linearized constraint manifold, justifying its use for enforcing both linear and nonlinear constraints.

**Proposition E.1** (Tangent-Space Projection in Hilbert Spaces)**.** *Let $\mathcal{H}$ be a real Hilbert space and let $h : \mathcal{H} \to \mathbb{R}^m$ be a Fréchet-differentiable constraint operator. Consider a point $u_1 \in \mathcal{H}$, and define the Jacobian $J := Dh(u_1) \in \mathcal{L}(\mathcal{H}, \mathbb{R}^m)$, the bounded linear operator representing the Fréchet derivative.*

*Then, the update*

$$u_{\text{proj}} = u_1 - J^*(JJ^*)^{-1}h(u_1)$$

*is the unique solution to the constrained minimization problem*

$$\min_{u \in \mathcal{H}} \|u - u_1\|_{\mathcal{H}}^2 \quad \textit{subject to} \quad h(u_1) + J(u - u_1) = 0,$$

*and corresponds to the orthogonal projection of $u_1$ onto the affine subspace defined by the linearization of $h$ at $u_1$, i.e., the tangent space to the constraint manifold at $u_1$.*

*In the special case where $h(u) = Au - b$ is affine, with $A \in \mathcal{L}(\mathcal{H}, \mathbb{R}^m)$, the update becomes the exact projection onto the feasible set $\{u \in \mathcal{H} : Au = b\}$.*

## F    Connection to ECI as a Special Case

The ECI (Extrapolate–Correct–Interpolate) framework [12] only considers and showcases empirical results on constraints such as fixed initial conditions, Dirichlet boundary values, and global mass integrals. These are all linear and non-overlapping, and are applied at isolated subsets of the solution field. To be used with the ECI framework, such constraints must admit closed-form oblique projections and act independently across disjoint regions. Let us denote any value or regional constraint [12] as a linear constraint $h(u) = Au - b$, for some matrix $A \in \mathbb{R}^{m \times n}$, vector $b \in \mathbb{R}^m$, and for simplicity and comparison purposes, the ECI hyperparameters such as re-sampling is set to its default value (0) and mixing to be 1.

In this regime, our Gauss–Newton update during PCFM guidance simplifies to an exact constraint satisfaction step for linear constraints:

$$u_{\text{proj}} = u_1 - A^\top (AA^\top)^{-1}(Au_1 - b),$$

which is precisely the "correction" step ECI employs. Moreover, by disabling the relaxed penalty ($\lambda = 0$) and skipping intermediate corrections, PCFM reduces exactly to ECI.

Hence, ECI is a special case of PCFM, tailored to simple linear constraints, while PCFM provides a general and principled framework for enforcing arbitrary nonlinear equality constraints during generative sampling.

## G    Pre-trained Models with Functional Flow Matching

### G.1    Flow Matching

Flow Matching (FM) [4, 36, 39] formulates generative modeling as learning a time-dependent velocity field $v_\theta(u, \tau)$ that transports a prior sample $u_0 \sim \pi_0$ to a target sample $u_1 \sim \pi_1$. The induced flow $\phi_\tau$ satisfies the ODE

$$\frac{d}{d\tau}\phi_\tau(u) = v_\theta(\phi_\tau(u), \tau), \quad \phi_0(u_0) = u_0,$$

and defines a trajectory $u(\tau) = \phi_\tau(u_0)$ connecting $\pi_0$ to $\pi_1$. The learning objective minimizes the squared deviation between the model field and a known target field $\hat{v}$, evaluated along interpolants $u_\tau = (1 - \tau)u_0 + \tau u_1$. This yields the standard Flow Matching loss:

$$\mathcal{L}_{\text{FM}}(\theta) = \mathbb{E}_{\tau \sim \mathcal{U}[0,1], \, u_0 \sim \pi_0, \, u_1 \sim \pi_1} \left[\|v_\theta(u_\tau, \tau) - \hat{v}(u_\tau, \tau)\|^2\right], \tag{27}$$

where $\hat{v}(u_\tau, \tau)$ is the true vector field, which is typically unknown.

**Conditional Flow Matching (CFM).**    When the conditional velocity field $u_t(u_1) := \partial_\tau \psi_\tau(u_1)$ is known in closed form, as under optimal transport, one can minimize the conditional flow matching loss:

$$\mathcal{L}_{\text{CFM}}(\theta) = \mathbb{E}_{\tau, u_0, u_1} \left[\|v_\theta(u_\tau, \tau) - (u_1 - u_0)\|^2\right], \tag{28}$$

where $u_\tau = (1 - \tau)u_0 + \tau u_1$ is a linear interpolant. This CFM objective forms the basis for most recent flow-based generative models due to its tractability and effectiveness.

## G.2 Functional Flow Matching

To handle infinite-dimensional generative tasks such as PDE solutions, Functional Flow Matching (FFM) [11] extends CFM to Hilbert spaces $\mathcal{U}$ of functions $u : \mathcal{X} \to \mathbb{R}$. The FFM flow $\psi_\tau$ satisfies

$$\frac{d}{d\tau}\psi_\tau(u) = v_\theta(\psi_\tau(u), \tau), \quad u_0 \sim \mu_0,$$

where $\mu_0$ is a base measure over function space. Under regularity assumptions, FFM minimizes an analogous loss over the path of measures:

$$\mathcal{L}_{\mathrm{FFM}}(\theta) = \mathbb{E}_{\tau, u_0, u_1}\left[\|v_\theta(u_\tau, \tau) - (u_1 - u_0)\|^2_{L^2(\mathcal{X})}\right], \tag{29}$$

where $u_\tau = (1 - \tau)u_0 + \tau u_1$ and the $L^2$-norm is used to evaluate function-valued velocities over the domain $\mathcal{X}$. This loss generalizes CFM to the functional setting, and serves as the pretraining objective for PCFM in this work.

# H Sampling Setups for PCFM and Other Baselines

For all comparisons, we adopt the explicit Euler integration scheme unless otherwise specified. We use 100 Euler update (flow matching) steps for heat and Navier-Stokes, and 200 steps for Reaction-Diffusion and Burgers (IC or BC).

**Vanilla FFM.** We perform unconstrained generation by integrating the learned flow vector field using explicit Euler steps, following the standard flow matching procedure [4, 11]. At each step, the model applies the forward update $u_{t+\Delta t} = u_t + \Delta t\, v_\theta(t, u_t)$ without any constraint enforcement.

**PCFM (Ours).** PCFM performs explicit Euler integration combined with constraint correction at every step. We apply 1 Newton update to project intermediate states onto the constraint manifold. Additionally, we optionally apply guided interpolation with $\lambda = 1.0$, step size 0.01, and 20 refinement steps to further refine interpolated states, although we use $\lambda = 0$ by default (no interpolation guidance) for computational efficiency (see Appendix L.3 for further ablations on $\lambda$). Interpolation guidance is only applied for the heat equation case to obtain results in Table 3. For Navier-Stokes, we follow the stochastic interpolant idea and randomize noise over batches [37] (i.e., different noise samples over batches). Finally, after all flow matching steps, we apply a final projection on the solution to enforce the constraints.

**ECI.** We follow the ECI sampling procedure introduced by Cheng et al. [12] where it performs iterative extrapolation-correction-interpolation steps as well as several mixing and noise resampling steps throughout the trajectory. For the simpler heat equation case, we did not do mixing and noise resampling. For more challenging PDEs (Navier-Stokes, Reaction-Diffusion, and Burgers), we apply $n_{\mathrm{mix}} = 5$ mixing updates per Euler step, and resample the Gaussian prior every 5 steps. We adopt their value enforcements in IC and Dirichlet BC cases, and also constant mass integral enforcement in the heat and Navier-Stokes problems, where the periodic boundary conditions lead to trivial linear constraints.

**D-Flow.** Following Ben-Hamu et al. [35], D-Flow differentiates through an unrolled Euler integration process. Following their paper and the set-up in Cheng et al. [12], we optimize the initial noise via LBFGS with 20 iterations and a learning rate of 1.0. The optimization minimizes the constraint violation at the final state, requiring gradient computation through the full unrolled trajectory. We adopt the adjoint method shipped from `torchdiffeq` [38] to differentiate through the ODE solver. Depending on the problem, we adopt an IC or BC loss as well as a PINN-loss based on the differential form of the PDE to form the constraint loss function to be optimized. We use 10 iterations and a learning rate of $10^{-2}$ for the Navier-Stokes dataset to avoid NaNs in the optimization loop.

**DiffusionPDE.** We adopt a gradient-guided sampling method proposed by Huang et al. [10] on the same pretrained FFM model. The generation process is augmented with explicit constraint correction where a composite loss of IC/BC and PINN [22] (differential form of the PDE), like in the D-Flow set-up, is used to guide the vector field at each flow step. We apply a correction coefficient $\eta = 1.0$ and set the PINN loss weight to $10^{-2}$ for stable generation. We find that a higher PINN weight or higher learning rate lead to unstable generation.

**PDM.** We implement the PDM algorithm proposed by Christopher et al. [49] for constrained generation. Although PDM also employs projection during sampling, its assumptions and modeling framework differ fundamentally from those of PCFM. For a fair comparison, we additionally implemented a PDM-style ablation within our flow models using the proposed projection step. While PDM achieves comparable performance on simpler systems such as the Heat equation, PCFM consistently outperforms it on more complex nonlinear dynamics, suggesting that PDM's strict per-step projections over-constrain the sampling trajectory and degrade generation quality.

# I  Batched Differentiable Solver for Nonlinear Constraints

We describe our solver for projecting a predicted sample $u_1 \in \mathbb{R}^n$ onto the nonlinear constraint manifold $\mathcal{M} := \{u \in \mathbb{R}^n : h(u) = 0\}$, where $h : \mathbb{R}^n \to \mathbb{R}^m$. The projection is formulated as the constrained optimization problem:

$$\min_{u \in \mathbb{R}^n} \ \tfrac{1}{2}\|u - u_1\|^2 \quad \text{subject to} \quad h(u) = 0. \tag{30}$$

The corresponding Lagrangian is:

$$\mathcal{L}(u, \lambda) = \tfrac{1}{2}\|u - u_1\|^2 + \lambda^\top h(u),$$

with first-order optimality conditions:

$$\nabla_u \mathcal{L}(u, \lambda) = u - u_1 + J(u)^\top \lambda = 0, \qquad h(u) = 0,$$

where $J(u) := \nabla h(u) \in \mathbb{R}^{m \times n}$ is the constraint Jacobian.

These yield the full nonlinear KKT system:

$$\begin{cases} u - u_1 + J(u)^\top \lambda = 0, \\ h(u) = 0. \end{cases} \tag{31}$$

**Newton-Based Update.** At iteration $k$, we linearize the KKT system around $u^{(k)}, \lambda^{(k)}$ and solve for updates $\delta u, \delta \lambda$. The full Newton step solves:

$$\begin{bmatrix} I + \sum_{i=1}^m \lambda_i^{(k)} \nabla^2 h_i(u^{(k)}) & J(u^{(k)})^\top \\ J(u^{(k)}) & 0 \end{bmatrix} \begin{bmatrix} \delta u \\ \delta \lambda \end{bmatrix} = - \begin{bmatrix} u^{(k)} - u_1 + J(u^{(k)})^\top \lambda^{(k)} \\ h(u^{(k)}) \end{bmatrix}. \tag{32}$$

Here, the upper-left block contains the Hessian of the Lagrangian:

$$\nabla^2_{uu} \mathcal{L}(u^{(k)}, \lambda^{(k)}) = I + \sum_{i=1}^m \lambda_i^{(k)} \nabla^2 h_i(u^{(k)}).$$

After solving, we update the primal and dual variables:

$$u^{(k+1)} = u^{(k)} + \delta u, \quad \lambda^{(k+1)} = \lambda^{(k)} + \delta \lambda.$$

This Newton system converges quadratically under standard regularity assumptions (e.g., full-rank Jacobian and Lipschitz-continuous second derivatives). In practice, we often omit the second-order terms to yield a Gauss–Newton approximation that is more stable and efficient in high-dimensional settings.

**Approximate KKT Solve via Schur Complement.** For inference-time projection, we adopt a simplified and batched update using the Schur complement [69]. At each iteration, we set $\lambda = 0$, and solve only for the primal update. Eliminating $\delta u$ from Equation (32), we obtain:

$$\left(J J^\top\right) \lambda = h(u), \qquad \delta u = -J^\top \lambda. \tag{33}$$

This gives the Gauss–Newton-style update:

$$u \leftarrow u_1 - J^\top (J J^\top)^{-1} \left(h(u) + J(u_1 - u)\right). \tag{34}$$

We iterate this procedure until convergence or until the constraint residual $\|h(u)\|$ falls below a set tolerance. The matrix $J J^\top \in \mathbb{R}^{m \times m}$ is small and typically well-conditioned for local or sparse constraints, enabling efficient solves.

Restarting the dual variable with $\lambda = 0$ at each iteration avoids stale gradient accumulation and improves numerical stability. This leads to a robust and memory-efficient projection routine that supports batched execution and reverse-mode differentiation.

**Batched and Differentiable Implementation.** We implement the solver in a batched and differentiable fashion to support inference across samples. For a batch of inputs $\{u_1^i\}_{i=1}^B$, we evaluate Jacobians $J^i$, residuals $h(u^i)$, and solve the corresponding Schur systems in parallel using vectorized operations and autodiff-compatible backends (e.g., PyTorch with batched Cholesky or linear solvers) [71]. Potentially, specific GPU kernels can be built for ODE integration to ensure optimal performance [38, 72].

**Computational Complexity.** The per-sample cost includes:

- $\mathcal{O}(m^2 n)$ for computing $J$ and $J^\top$,
- $\mathcal{O}(m^3)$ for solving the Schur complement system,
- $\mathcal{O}(n)$ for applying the update.

Since typically $m \ll n$, the overall cost scales as $\mathcal{O}(n)$ per sample.

## J  Evaluation Metrics

We evaluate each method using 512 generated samples for all 1D problems and 100 samples for the 2D Navier-Stokes problem. For each setting, we use an equivalent number of ground truth PDE solutions with a fixed IC or BC configuration used during sampling for direct comparison.

**MMSE and SMSE.** Following Kerrigan et al. [11], Cheng et al. [12], we evaluate generation fidelity with mean of the MSE and the standard deviation of the MSE as:

$$\text{MMSE} = \|\mu_{\text{gen}} - \mu_{\text{gt}}\|_2^2, \quad \text{SMSE} = \|\sigma_{\text{gen}} - \sigma_{\text{gt}}\|_2^2,$$

where $\mu_{\text{gen}}, \mu_{\text{gt}}$ denote the mean across generated and true PDE solutions, and $\sigma_{\text{gen}}, \sigma_{\text{gt}}$ the standard deviations.

**Constraint Error.** To evaluate physical consistency, we compute the $\ell_2$ norm of residuals from constraint functions $\mathcal{R}_*$ applied to each sample $\hat{u}^{(n)}$, then average across all $N$ samples:

$$\text{CE}(*) = \frac{1}{N} \sum_{n=1}^N \left\| \mathcal{R}_* \left( \hat{u}^{(n)} \right) \right\|_2, \quad * \in \{\text{IC}, \text{BC}, \text{CL}\}.$$

where the residuals are computed following Appendix D to measure the constraint or conservation violation.

**Fréchet Poseidon Distance (FPD).** To assess distributional similarity beyond mean and variance, we adopt the Fréchet Poseidon Distance (FPD) introduced in Cheng et al. [12], where we measure the Fréchet distance between the hidden state distributions extracted from a pretrained foundation model (Poseidon [67]) applied to both generated and true solutions. We pass both generated and true solutions through the Poseidon base model (157M parameters) and extract the last hidden activations of the encoder to eventually obtain a 784-dimension feature vector for FPD calculation:

$$\text{FPD}^2 = \|\mu_1 - \mu_2\|^2 + \text{Tr}\left( \Sigma_1 + \Sigma_2 - 2 \left(\Sigma_1 \Sigma_2\right)^{1/2} \right), \tag{35}$$

where $\mu_1, \Sigma_1$ and $\mu_2, \Sigma_2$ are the empirical mean and covariance of the Poseidon embeddings from the generated and true solutions' distributions, respectively. FPD is computed either per frame $u(x, y)$ for 2D PDEs and averaged over all frames (across $t$) or over the full spatiotemporal solution $u(x, t)$ for 1D PDEs.

# K   Further Results

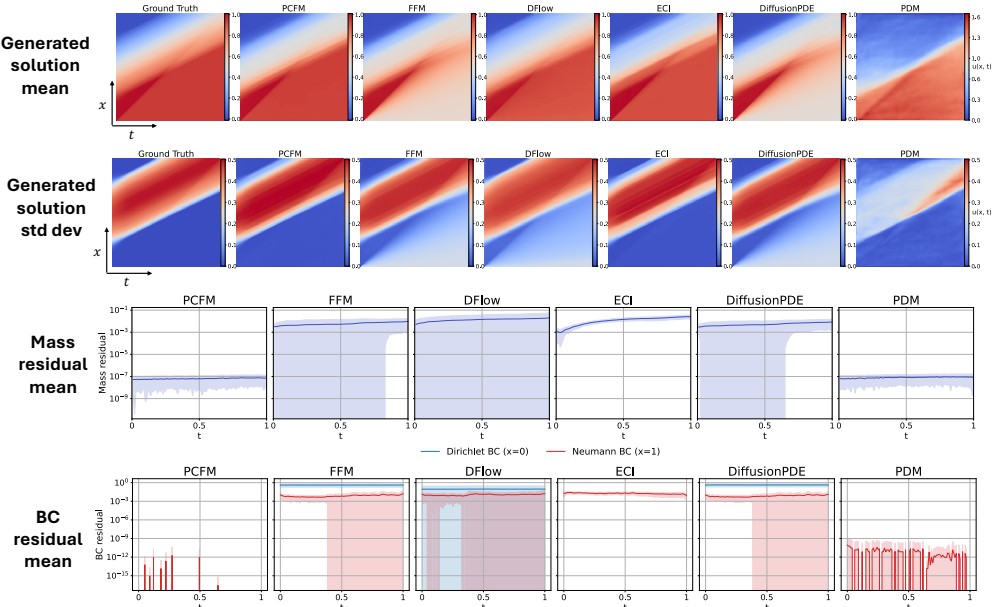

Figure 5: Solution profiles for the Inviscid Burgers equation with fixed BC. We plot the various constraint guidance methods and compare the mean solution profile and standard deviation. While PCFM yields slightly worse MMSE and SMSE and better FPD, it ensures global mass conservation and maintains low constraint errors for both Dirichlet and Neumann BCs over time.

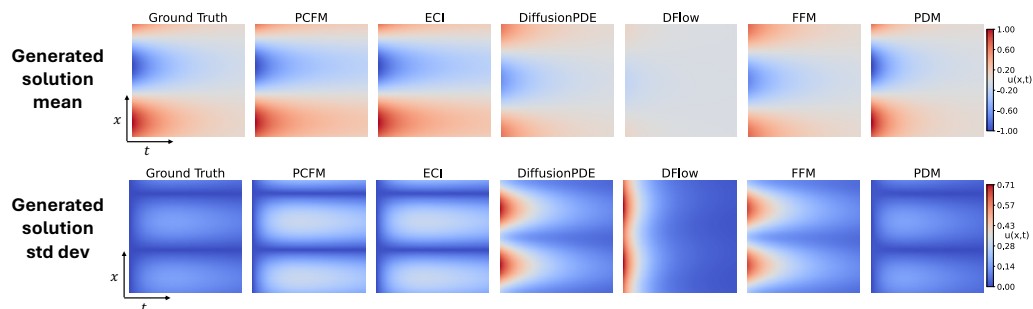

Figure 6: Solution profiles for the Heat equation with fixed IC. We plot the various constraint guidance methods and compare the mean solution profile and standard deviation. PCFM outperforms all other methods by visually being the most similar to the ground truth.

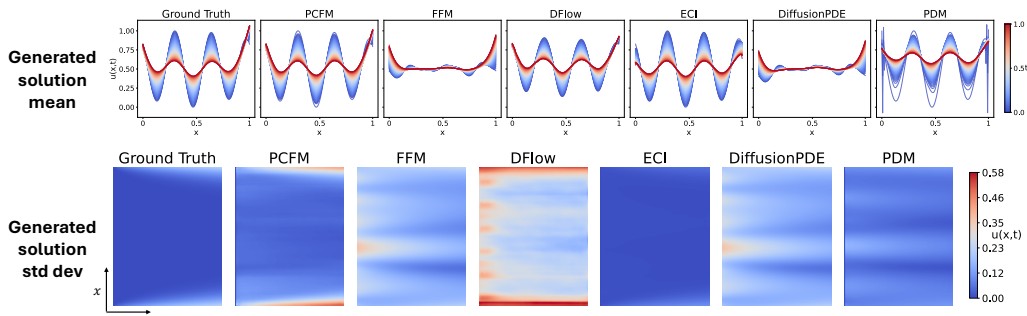

Figure 7: Alternative view of the Reaction-Diffusion equation with fixed IC. We plot the pointwise mean and standard deviation across generated samples for each method. PCFM outperforms all other methods by visually being the most similar to the ground truth.

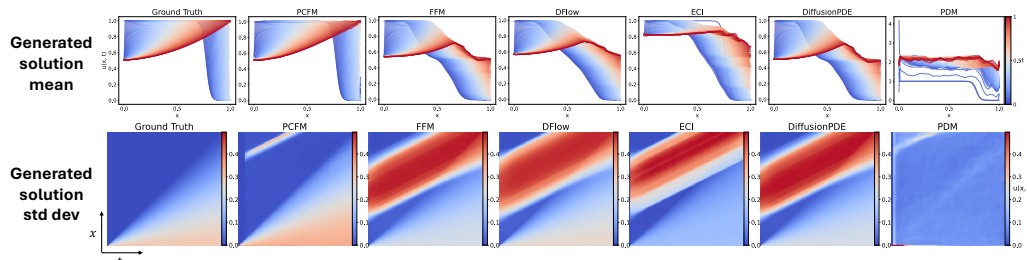

Figure 8: Alternative view of the Inviscid Burgers equation with fixed IC. We plot the pointwise mean and standard deviation across generated samples for each method. PCFM outperforms all other methods by visually being the most similar to the ground truth.

## L    Further Ablations

### L.1    Effect of Intermediate Projections

Physical constraints such as mass conservation or initial conditions are typically imposed only on the final state. However, we find that applying projections at intermediate steps improves flow consistency and numerical stability. This parallels practices in diffusion and flow-matching models, where intermediate states, often initialized as noise, need not satisfy constraints or carry semantic meaning.

To assess this, we perform an ablation presented in Table 6, in which projections are applied only at the final step, omitting intermediate corrections. We observe that trajectories drift farther from the constraint manifold, resulting in larger corrective updates, slower convergence, and lower accuracy.

Intermediate projections, by contrast, act as continuous guidance, progressively steering samples toward the feasible manifold while preserving diversity. This avoids over-constraining noisy intermediates and yields more accurate and stable final solutions.

### L.2    Total Variation (TV) Constraints on the Heat Equation

Total Variation Diminishing (TVD) constraints encode the principle that diffusive systems, such as those governed by the heat equation, smooth spatial fluctuations over time. Mathematically, this is expressed by the fact that the total variation of the solution should not increase in time:

$$\text{TV}(u(t)) := \int \left| \frac{\partial u}{\partial x} \right| dx \quad \text{is non-increasing in } t.$$

To enforce this property in a data-driven or generative setting, we adopt a hard constraint that relates the total variation at final time $T$ to the total variation at the initial time $t = 0$. Specifically, we

Table 6: Ablation study on intermediate projections. Applying projections only at the final step leads to larger constraint deviations and higher errors, while PCFM's intermediate corrections improve stability and accuracy across all datasets.

| Dataset | Method | MMSE/ $10^{-2}$ | SMSE/ $10^{-2}$ | CE (IC) | CE (CL) |
|---|---|---|---|---|---|
| Heat | Project final only | 4.31 | 3.27 | **0** | **0** |
| | PCFM | **0.241** | **0.937** | **0** | **0** |
| Burgers IC | Project final only | 10.59 | 6.90 | **0** | 4.05 |
| | PCFM | **0.052** | **0.272** | **0** | **0** |
| Reaction–Diffusion | Project final only | 2.82 | 2.48 | **0** | **0** |
| | PCFM | **0.026** | **0.583** | **0** | **0** |
| Navier–Stokes | Project final only | 16.47 | 8.60 | **0** | **0** |
| | PCFM | **4.59** | **4.17** | **0** | **0** |

impose the condition:
$$\mathrm{TV}(u_T) = \gamma \cdot \mathrm{TV}(u_0),$$
where $\gamma \in (0, 1)$ is a decay factor that can either be fixed (e.g., $\gamma = 0.5$) or estimated from unprojected samples. This constraint can be implemented using discrete spatial differences as:
$$\mathrm{TV}(u_j) \approx \sum_{i=0}^{n_x-2} |u_{i+1,j} - u_{i,j}|,$$
applied at selected time slices (typically $t = 0$ and $t = T$).

We encode the constraint as a differentiable function:
$$h_{\mathrm{TV}}(u) := \mathrm{TV}(u_T) - \gamma \cdot \mathrm{TV}(u_0) = 0.$$

This constraint is compatible with mass conservation and initial condition enforcement, and can be combined with other physically informed conditions such as energy decay or curvature regularity. In practice, we observe that enforcing TVD constraints reduces spurious oscillations in the generative output and improves structural fidelity without significantly altering statistical accuracy.

**Power MSE.** To further quantify the effect of this constraint, we introduce the *Power MSE* metric, which measures the deviation in spatial frequency content between the generated and ground truth solutions. Formally, for a model's mean output $\bar{u}(x, t)$, we compute the spatial power spectrum $|\hat{u}(k, t)|^2$ and average over time to obtain a per-frequency energy profile. The Power MSE is then defined as:
$$\text{Power MSE} := \frac{1}{n_x} \sum_k \left( \overline{|\hat{u}_{\mathrm{gen}}(k)|^2} - \overline{|\hat{u}_{\mathrm{ref}}(k)|^2} \right)^2.$$

This metric captures discrepancies in frequency modes and is especially sensitive to unphysical sharpness or over-smoothing. Variants of power-spectrum-based error metrics have been used in turbulence modeling [73], climate downscaling [74], and Fourier neural operators training [23]. In our setting, we find that Power MSE effectively complements pointwise metrics (e.g., MMSE, CE), as it evaluates structural fidelity in the spectral domain. As shown in Table 7, this metric reveals how enforcing a TVD constraint not only improves calibration but also leads to closer alignment in frequency space.

**Effect of TVD Constraint.** To isolate the benefit of total variation control, we mirror the **ECI** setting by setting the constraint penalty weight to zero and enforcing only initial condition and mass conservation constraints. When we additionally include the TVD constraint ($\gamma = 0.3$) in our **PCFM** model (with no penalty-based regularization), we observe a consistent improvement in structural fidelity. This is most evident in the Power MSE metric, where PCFM achieves the lowest error among all baselines (488.42 vs. 500.12 for ECI), indicating better alignment with the true spatial energy distribution. We also see gains in MMSE and SMSE, without introducing constraint violations (e.g., CE remains zero). This confirms that the TVD constraint acts as a meaningful structural prior even in the absence of learned penalties, improving generative realism without sacrificing calibration or efficiency.

Table 7: Test metrics on the Heat Equation dataset with TVD constraints. Lower is better for all metrics.

| Metric | PCFM | ECI | DiffusionPDE | D-Flow | FFM |
|---|---|---|---|---|---|
| MMSE / $10^{-2}$ | **0.684** | 0.697 | 4.49 | 1.97 | 4.56 |
| SMSE / $10^{-2}$ | **0.962** | 0.973 | 3.93 | 1.14 | 3.51 |
| CE (IC) / $10^{-2}$ | **0** | **0** | 599 | 102 | 579 |
| CE (CL) / $10^{-2}$ | **0** | **0** | 2.06 | 64.8 | 2.11 |
| FPD | **1.28** | 1.34 | 1.70 | 2.70 | 1.77 |
| *Power MSE* | **488.42** | 500.12 | 1420.92 | 3996.11 | 932.20 |

## L.3 Effect of Relaxed Constraint Correction with Flow Matching Steps

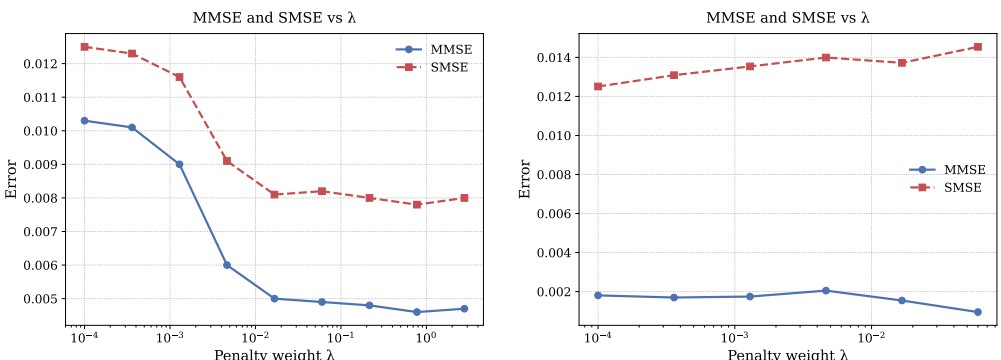

Figure 9: Effect of penalty weight $\lambda$ on MMSE and SMSE for the Reaction-Diffusion dataset. Left: 10 flow matching steps. Right: 100 flow matching steps.

Figure 9 shows the effect of the relaxed constraint penalty weight $\lambda$ on the generation error, measured via MMSE and SMSE, for the Reaction-Diffusion dataset. We evaluate on a $128 \times 100$ spatial-temporal grid, as described in Appendix B.3, and solve the relaxed optimization objective in Equation (6) using the Adam optimizer.

With only 10 flow steps (left), increasing $\lambda$ significantly improves performance, as the relaxed constraint correction effectively compensates for the coarser integration of the flow. In contrast, with 100 flow steps (right), the flow is more precise, and additional penalty yields only marginal benefit. This illustrates that relaxed correction is particularly valuable when inference is constrained to a small number of steps—for instance, in scenarios where evaluating the vector field $v_\theta$ is computationally expensive.

However, overly large values of $\lambda$ can harm performance by distorting the reverse update and breaking alignment with the OT interpolant (see Proposition 3.1), as discussed in Equation (5). Hence, choosing $\lambda$ requires balancing constraint enforcement with consistency along the learned generative path.

## L.4 Evaluation on Generating Out-of-Distribution Samples

A key advantage of inference-time constraint enforcement is its ability to generalize beyond the constraints implicitly present in the training data. To assess this capability, we evaluate PCFM under *out-of-distribution* (OOD) constraint settings, where the pretrained model encounters constraint realizations that were never observed during training. Specifically, we design an OOD experiment based on the Burgers equation, focusing on shifts in the initial condition (IC). The pretrained flow-matching model is trained on a standard dataset in which the initial sigmoid profile

$$u(x,0) = \frac{1}{1 + \exp[-\alpha(x - x_0)]}$$

is centered at locations $x_0 \in [0.4, 0.6]$. For testing, we construct a new OOD dataset where $x_0$ is sampled from the non-overlapping intervals $x_0 \in [0.0, 0.2]$ and $x_0 \in [0.8, 1.0]$. This shift introduces

Table 8: Performance on out-of-distribution (OOD) dataset for the Burgers equation. PCFM achieves the lowest errors while exactly satisfying both initial-condition (IC) and conservation-law (CL) constraints, demonstrating robust generalization beyond training-distribution constraints.

| Method | MMSE / $10^{-2}$ | SMSE / $10^{-2}$ | CE (IC) | CE (CL) |
|---|---|---|---|---|
| PCFM | **1.83** | **1.40** | **0** | **0** |
| ECI | 13.30 | 5.65 | **0** | 3.64 |
| D-Flow | 15.59 | 7.82 | 6.00 | 0.092 |
| Vanilla | 18.84 | 7.88 | 6.37 | 0.069 |
| DiffusionPDE | 19.49 | 8.03 | 6.42 | 0.062 |
| PDM | 206.6 | 2.81 | **0** | **0** |

initial conditions that lie outside the training distribution, thereby evaluating each method's ability to adapt to unseen constraint configurations.

All models are evaluated in a zero-shot setting, reusing the pretrained FFM backbone without retraining. We employ 200 flow-matching steps for the Burgers problem to ensure stable integration and consistent comparison across methods. Each approach enforces the same IC and conservation constraints, providing a fair assessment of generalization ability.

The results, summarized in Table 8, show that PCFM achieves the lowest mean and sample-wise mean squared errors (MMSE and SMSE) while exactly satisfying both IC and conservation-law constraints, even under this challenging OOD regime. In contrast, unconstrained or partially constrained baselines exhibit significant violations, confirming that inference-time projection enables robust generalization to unseen constraint configurations.

