# OpenReview forum: "Physics-Constrained Flow Matching: Sampling Generative Models with Hard Constraints"
_NeurIPS.cc/2025/Conference — NeurIPS 2025 poster_

### Official Review · Reviewer_5iKF · 2025-06-26

**Clarity:** 3
**Significance:** 4
**Originality:** 2
**Rating:** 5
**Confidence:** 2

**Summary:**

This paper proposes a method to guide the inference-time generation process of a flow-based vector field model $v_\theta$, which has been trained to simulate the behavior of physical systems governed by partial differential equations (PDEs), so that the outputs satisfy physical constraints such as initial/boundary conditions and conservation laws (both linear and nonlinear), without requiring retraining of $v_\theta$.
To generate the state $u_\tau$ for $\tau \in [0,1]$, the method first solves an ODE based on $v_\theta$ forward from $u_\tau$ to estimate $u_1$. The estimated $u_1$ is then projected onto the manifold defined by the given constraints using a Gauss–Newton projection. An ODE is then solved backward from the projected $u_1$ to determine a correction target for $u_\tau$. Finally, a penalized correction (Eq. (6)) based on this target is applied to obtain the final corrected $u_\tau$.

The proposed method is applied to functional flow models (FFMs) trained to simulate physical systems such as the Navier–Stokes and Burgers’ equations. The results show that the method produces outputs that numerically satisfy the imposed constraints with high accuracy. In contrast to existing approaches, which often struggle to strictly enforce nonlinear constraints, the proposed method successfully generates solutions that comply with them. Furthermore, enforcing these constraints improves performance on evaluation metrics such as mean squared error (MSE) and FPD.

**Questions:**

### Comments and Typos
It seems that an additional operation such as $u \leftarrow u_{\tau'}$ may be required after line 8 in Algorithm 1.

Algorithm 1 should be explicitly referenced and discussed in the main text.

Line 127: constrained -> Constrained?

Line 136: $\mathcal{F}(u; \phi)$ -> $\mathcal{F}_{\phi}(u)$?

**Ethical Concerns:**

["NO or VERY MINOR ethics concerns only"]

**Final Justification:**

The authors' response clarified several aspects of the proposed method. While the technical novelty may be somewhat limited given its similarity to ECI, I find that the paper still makes a meaningful contribution. In particular, the experimental results demonstrate that PCFM handles nonlinear constraints more efficiently than existing methods, which I consider a valuable empirical finding.

I have no major remaining concerns, and I maintain my original score in support of acceptance.

**Limitations:**

See weaknesses above.

**Paper Formatting Concerns:**

No concerns.

**Quality:**

3

**Strengths And Weaknesses:**

### Strengths
* The proposed method successfully imposes both linear and nonlinear physical constraints on flow-based generative models of physical systems in a zero-shot manner.
* The experiments demonstrate that the proposed method can generate solutions that satisfy the constraints, and that enforcing such constraints also improves the accuracy of the generated results.

### Weaknesses
* The proposed method shares a broadly similar procedure with the existing ECI framework, which limits its technical novelty. As far as I understand, the primary distinction lies in the correction step introduced in Eq. (6).
* The experiments are limited to relatively small-scale settings with only two degrees of freedom.

---

> ### Author Rebuttal · Authors · 2025-07-31
>
> We thank the reviewer for their thoughtful and constructive review. We appreciate your recognition of the method’s ability to enforce both linear and nonlinear physical constraints on flow-based generative models in a **zero-shot** manner and your general positive inclination towards our method. Below, we address the specific concerns raised.
>
> ## R4-1: Novelty Relative to the ECI Framework
>
> We appreciate the reviewer’s observation regarding the similarities to the ECI framework. While our method shares a conceptual goal: enforcing constraints at inference time, it differs significantly in both **generality** and **mechanism**, and we clarify these distinctions below:
>
> * **Generality of Constraints**: ECI introduces a novel mixing-based correction strategy but is primarily demonstrated on **simple, linear, and decoupled constraints**. It does not provide a general framework for handling **nonlinear, coupled, or global constraints**, nor does it show robust performance on PDEs with **shocks or discontinuities**. In contrast, PCFM is **built to support arbitrary nonlinear equality constraints**, including **nonlocal mass conservation**, and shows strong empirical performance in such regimes.
>
> * **Theoretical Connection**: As detailed in Appendix F, ECI is a **special case** of PCFM when constraints are linear, non-overlapping, and the penalty is omitted (λ = 0). PCFM generalizes this setting to support more **complex and coupled constraints**.
>
> * **Extensibility and Differentiability**: Although ECI is described as “gradient-free,” it depends on **analytical projections** that do not extend naturally to nonlinear or global constraints. PCFM uses **general-purpose automatic differentiation** and **Jacobian-vector products**, enabling scalable enforcement in **high-dimensional and nonlinear settings**.
>
> * **Inference Strategy**: ECI corrects the final output through constrained optimization, which can be expensive in large systems. PCFM instead uses **efficient projection steps during generation** and applies a **single correction step** at the end (Eq. 6) to ensure feasibility without high overhead.
>
> We agree that Eq. 6 is a key contribution: it ensures feasibility while promoting alignment with the learned flow field, preserving generative consistency. We validate this in Appendix L.2, where PCFM outperforms ECI even under linear constraints, and more strongly under nonlinear and coupled constraints.
>
> That said, Eq. 6 is part of a broader pipeline. PCFM integrates projection-based solvers, reversible integration, and constraint-guided updates, drawing from numerical PDE solvers to enable general, zero-shot constraint enforcement. Unlike ECI, which solves a constrained optimization problem at every sampling step, PCFM uses lightweight intermediate projections and a final correction. To ensure scalability and efficiency, we implement a batched, differentiable projection solver using reverse-mode Automatic Differentiation and Schur-complement updates, avoiding retraining while supporting complex constraints.
>
> ## R4-2: Experimental Scope — Small-Scale Systems
>
> We respectfully clarify that while the reviewer refers to our setting as having “only two degrees of freedom,” the Navier–Stokes system is inherently three-dimensional, spanning two spatial dimensions and one temporal dimension. Moreover, we enforce three types of constraints simultaneously—initial conditions, local consistency, and nonlinear global conservation laws—highlighting the method’s applicability to high-dimensional and tightly constrained PDE systems.
>
> To ensure thorough evaluation, we included PDEs of varying difficulty: from relatively simple systems like the Heat equation, to intermediate ones such as Reaction–Diffusion, and finally to nonlinear, shock-prone systems like Inviscid Burgers and Navier–Stokes.
>
> All datasets and spatiotemporal resolutions are sourced directly from prior work (Appendix B). Evaluating on 2D and 3D PDE systems is standard practice in the literature on neural operator learning [1, 2], scientific foundation models [3, 4], and constrained generative modeling [5, 6], making our experimental setup both representative and widely accepted.
>
>
> [1] Li, Zongyi, et al. "Fourier Neural Operator for Parametric Partial Differential Equations." NeurIPS (2020).
>
> [2] Li, Zongyi, et al. "Physics-Informed Neural Operator for Learning Partial Differential Equations." arXiv:2111.03794 (2021).
>
> [3] Herde, Maximilian, et al. "Poseidon: Efficient Foundation Models for PDEs." arXiv:2405.19101 (2024).
>
> [4] Sun, Jingmin, et al. "Towards a Foundation Model for Partial Differential Equation: Multi-Operator Learning and Extrapolation." arXiv:2404.12355 (2024).
>
> [5] Huang, Jiahe, et al. "DiffusionPDE: Generative PDE-Solving under Partial Observation." arXiv:2406.17763 (2024).
>
> [6] Négiar, Geoffrey, Michael W. Mahoney, and Aditi S. Krishnapriyan. "Learning Differentiable Solvers for Systems with Hard Constraints." arXiv:2207.08675 (2022).
>
> Thank you for pointing out other minor presentation issues. In the revised manuscript, we will fix those typographical errors.

---

> > ### Comment · Reviewer_5iKF · 2025-08-04
> >
> > Thank you for the clarification.
> >
> > It remains unclear to me what exactly is meant by the “constrained optimization problem” in the phrase “ECI, which solves a constrained optimization problem at every sampling step.” However, the experimental results do indicate that PCFM handles constraints—particularly nonlinear ones—more efficiently than ECI. Based on this, I would prefer to maintain my current score.

---

> > > ### Author Response · Authors · 2025-08-05
> > >
> > > Thank you for your thoughtful follow-up.
> > >
> > > To clarify the statement that "ECI solves a constrained optimization problem at every sampling step": in ECI, each sample must be individually corrected via an explicit constrained optimization of the form
> > >
> > > $$
> > > \arg\min_{u} \||u - \hat{u}\||^2 \quad \text{s.t.} \quad h(u) = 0,
> > > $$
> > >
> > > which can be computationally expensive or restrictive when constraints are nonlinear or coupled.
> > >
> > > In contrast, **PCFM does *not* solve a full constrained optimization problem during sampling**. Instead, it uses a **lightweight Gauss–Newton update**
> > >
> > > $$
> > > u_{\text{proj}} \leftarrow u_1 - J^\top (JJ^\top)^{-1} h(u_1), \quad \text{where } J = \nabla h(u_1),
> > > $$
> > >
> > > as a **continuous guidance mechanism** to gently steer the trajectory toward feasibility. This update introduces minimal computational overhead and preserves compatibility with the flow-based generation process.
> > >
> > > Once sampling completes, a **constrained optimization problem** solve is performed **only if needed**, ensuring final constraint satisfaction.
> > >
> > > In addition, **Equation (6)** in the paper introduces a **penalized correction** that softly aligns the final state with the projected flow trajectory while promoting consistency with the learned dynamics. This correction acts as a principled way of promoting alignment with the learned flow field, preserving generative consistency and coherence without retraining, especially crucial when dealing with nonlinear constraints.
> > >
> > > We hope this clarifies the distinction.

---

> > > > ### Comment · Reviewer_5iKF · 2025-08-05
> > > >
> > > > Thank you for your response.
> > > >
> > > > According to the ECI paper (though I have not read it in full detail, so my understanding may be incomplete), it is true that ECI does not handle nonlinear constraints. However, ECI is based on the use of oblique projection to satisfy linear constraints, and from that perspective, I am not fully convinced that it is more computationally expensive than PCFM for the linear case. As noted in your Appendix F, the Gauss–Newton update of PCFM reduces to the same correction as ECI when the constraints are linear.
> > > >
> > > > That said, I agree that PCFM has a practical advantage in that it allows projection via automatic differentiation as long as $h(u)$ is defined, whereas ECI requires analytical projections, even in the linear case.

---

> > > > > ### Author Response · Authors · 2025-08-06
> > > > >
> > > > > Thank you for the clarification and for engaging further with the distinction between PCFM and ECI.
> > > > >
> > > > > You're absolutely right that in the linear case, both ECI and PCFM reduce to the same oblique projection step—this is precisely what we intended to convey in Appendix F. Our goal was not to claim computational superiority over ECI in the linear setting, but rather to highlight that **PCFM generalizes the projection mechanism to nonlinear, coupled, and global constraints** using automatic differentiation and Gauss–Newton updates, and would be computationally better in the nonlinear case.
> > > > >
> > > > > As you noted, this offers a **practical advantage**: PCFM applies to any differentiable constraint function $h(u)$, without requiring handcrafted analytic projections.
> > > > >
> > > > > Thank you again for your thoughtful review and engagement throughout the discussion.
> > > > >
> > > > > Best regards,
> > > > >
> > > > > The Authors

---

### Official Review · Reviewer_FfpW · 2025-07-01

**Clarity:** 3
**Significance:** 2
**Originality:** 2
**Rating:** 4
**Confidence:** 4

**Summary:**

This paper presents Physics-Constrained Flow Matching, a method that enables arbitrary constraint enforcement on generative flow matching outputs through inference time constraint imposition. PCFM applies constraints by projecting the estimated final state $u_1$ (which is computed using a forward ODE solver) and then reversing the ODE to compute the next state $u_{\tau'}$. As evaluated on a series of experiments evaluating performance on challenging PDE dynamics, PCFM outperforms baselines in qualitative metrics while ensuring perfect constraint adherence.

**Questions:**

Questions:
1. Has any analysis been taken for out-of-distribution constraints (e.g., constraints that aren't necessarily satisfied by the training data)?
2. Can the authors elaborate further on the SMSE performs worse as the number of flow matching steps increases? It seems intuitive that the MMSE decreases as the number of steps increases, but I'm having trouble reasoning as to its impact on SMSE.

**Ethical Concerns:**

["NO or VERY MINOR ethics concerns only"]

**Final Justification:**

The authors have successfully addressed my primary concerns during the rebuttal period. To summarize:

- **Comparison to ECI and PDM:** The additional results and explanations provided have addressed my primary concerns. Their inclusion in the final manuscript will mitigate this issue. While I believe there are some remaining questions of novelty (as the authors even state that their work is a generalized version of ECI), these additional points assist in differentiating the work from baselines.

- **Runtime:** This should be acknowledged in the limitations. The table provided should be added to the paper. It's inclusion addresses this concern.

- **Generality Claims:** The authors state their intent to clarify this and soften the claims in the updated manuscript. This will address my concern.

- **Out-of-Distribution Settings:** I believe this is interesting, and its inclusion would further strengthen the analysis.

Overall, while I still maintain some concerns about novelty, I have raised my score from a 3 to a 4 as I believe these clarifications push the paper over the acceptance threshold.

**Limitations:**

The primary limitation that seems to be missing is an explanation of the impact of PCFM on runtime. It would be helpful if the authors could comment on this point.

**Paper Formatting Concerns:**

No formatting concerns.

**Quality:**

3

**Strengths And Weaknesses:**

Strengths:
1. **General Applicability (Significance):** The method provides several important selling points: *(1) Arbitrary Constraint Enforcement:* PCFM can handle arbitrarily complex constraint sets, a property which is demonstrated on complex PDE constraint enforcement, *(2) Inference-Time Method:* PCFM does not require retraining, so can be easily adopted by existing pretrained models.
2. **Strong Empirical Performance (Significance):** PCFM reports strong results on the settings explored. The experimental settings are interesting, including varying degrees of constraint complexity on several different settings.
3. **Writing and Presentation (Clarity):** The paper is well written and easy to follow. The method is presented clearly and well described.

Weaknesses:
1. **Inadequate Comparison to Similar Work (Originality, Quality):** The use of intermediate projections for continuous-time generative models is not a foreign concept in constrained generation literature [1-2]. For example, [2] applies projections to intermediate states in the diffusion reverse process, providing many of the same benefits that PCFM boasts (e.g., arbitrary constraint satisfaction, zero-shot constraint imposition). Furthermore, as a projection operator has already been designed for the PCFM's inference, it would be simple to add [2] as a baseline in the experiments. Due to the overlap between the methodology in these papers, a comparison would help clarify the novelty and performance claims.
2. **Runtime Analysis (Significance):** PCFM relies on additional flow matching steps to balance constraint correction and sample quality. This (along with the complexity of computing the projection operator) likely contribute to much higher runtimes. Additional analysis should be included comparing the runtime of PCFM to the baselines. While the analysis in Appendix L.2 is interesting, it could be further expanded to include these computational overhead benchmarks as well.
3. **Generality of PCFM (Significance):** The experimental analysis focuses on synthetic data for PDE governed dynamics. While synthetic data is generally sufficient for papers focusing on PDEs [3-4], PCFM is presented as a general method (e.g. lines 14-16, lines 67-71) for constraint satisfaction in other settings as well. These claims would be better supported by including "real-world" experiments to supplement the synthetic settings examined.


---

[1] Cheng, Chaoran, et al. "Gradient-free generation for hard-constrained systems." (2025).

[2] Christopher, Jacob K., Stephen Baek, and Nando Fioretto. "Constrained synthesis with projected diffusion models." Advances in Neural Information Processing Systems 37 (2024): 89307-89333.

[3] Huang, Jiahe, et al. "DiffusionPDE: Generative PDE-solving under partial observation." arXiv preprint arXiv:2406.17763 (2024).

[4] Ben-Hamu, Heli, et al. "D-flow: Differentiating through flows for controlled generation." arXiv preprint arXiv:2402.14017 (2024).

---

> ### Author Rebuttal · Authors · 2025-07-31
>
> We thank the reviewer for the thoughtful assessment, noting the generality of our method and clarity of presentation. We appreciate recognition of PCFM’s handling of complex constraints at inference and strong empirical performance on challenging PDE benchmarks.
>
> ## R3-1: Comparison to Prior Work on Intermediate Projection
>
> While PCFM shares inference-time constraint satisfaction goals, it differs from ECI and Projected Diffusion Models (PDM) in generality, theory, and extensibility.
>
> ### Comparison to ECI:
>
> 1. ECI is a special case of PCFM (Appendix F), limited to linear, uncoupled constraints without correction penalty (λ = 0). PCFM generalizes this to support nonlinear and coupled constraints, such as nonlocal mass conservation.
>
> 2. ECI is tested on simple linear cases and lacks scalability for nonlinear/global constraints. Its correction involves a full constrained optimization, which is expensive in large systems. PCFM uses lightweight projections during generation for guidance and final correction for feasibility.
>
> 3. Though described as "gradient-free", ECI relies on analytical projections, limiting extensibility as robust optimization methods are gradient-based methods, which makes the extension to nonlinear constraints nontrivial. PCFM uses our custom batched differentiable solver for scalable PDE enforcement.
>
> ### Comparison with PDM
>
> While PDM also uses projection during sampling, its assumptions and modeling context differ fundamentally from PCFM.
>
> 1. PDM enforces constraints at every timestep, which can over-constrain the trajectory, especially under nonlinear or global constraints like ICs or mass conservation that only apply at the final state. PCFM avoids this by guiding the trajectory through reversible integration and applying a single final correction, preserving consistency and flexibility.
>
> 2. PDM is built on stochastic diffusion models, while PCFM uses deterministic flow matching. Though related via the probability flow ODE \[10], their sampling behaviors and practical implementation differ.
>
> 3. PDM has not been evaluated on PDEs with shocks or discontinuities, and its scalability to nonlinear global constraints is unclear. In contrast, PCFM is designed for such regimes and shows strong performance on complex PDEs like Burgers and Navier–Stokes.
>
> To ensure fair comparison, we implemented a PDM-style ablation on our flow models. While it performs comparably on simpler cases like Heat, PCFM clearly outperforms on harder nonlinear dynamics, validating our design choices.
>
> ### Table A: Comparison between PDM and PCFM
> | Dataset          | Method | MMSE      | SMSE      | CE (IC)      | CE (CL)  |
> |------------------|--------|-----------|-----------|--------------|----------|
> | Heat             | PDM    | 0.0045    | **0.0002**   | 0   | 0|
> | Heat    | PCFM      |    **0.00241**    |   0.00937     |         0    |       0       |
> | Burgers          | PDM    | 1.5309    | 0.0262    | 0  | 0|
> | Burgers | PCFM      |   **0.00052**        |     **0.00272**      |      0       |      0         |
> | Reaction-Diffusion | PDM    | 0.0174    | **0.0043**    | 0   | 0|
> | Reaction-Diffusion      | PCFM      |    **0.00026**       |     **0.00583**      |     0       |        0       |
> | Navier-Stokes    | PDM    | 0.1395    | 0.0786    | 0  | 0|
> | Navier-Stokes    | PCFM   | **0.0459**    | **0.0417**    | 0            | 0        |
>
>
> ## R3-2: Runtime Analysis and Overhead
>
> We agree that runtime is important for assessing PCFM’s practicality. Accordingly, we include detailed runtime and memory comparisons using consistent batch sizes (8 for Navier–Stokes, 32 otherwise) and 100 flow steps on a single NVIDIA V100 GPU.
>
> PCFM incurs a modest overhead relative to unconstrained baselines like vanilla FFM, due to exact constraint enforcement. However, it achieves 4–6× speedups over DFlow \[4], which requires costly backpropagation through the full ODE unroll. DiffusionPDE is faster but struggles with hard constraints, especially for nonlinear problems like Burgers and Reaction-Diffusion. ECI’s runtime grows with mixing steps but still fails to satisfy nonlinear constraints.
>
> For linear problems like Heat and Navier–Stokes, projections converge quickly, reflected in PDM’s runtime. Overall, PCFM adds a small, necessary overhead for robust constraint enforcement. We will summarize these findings in the main text.
>
> ### Table B: Runtime and Memory Comparison
>
> | **Dataset**          | **Metric**       | **PCFM** | **ECI** | **DiffusionPDE** | **D-Flow** |  **PDM**  | **Vanilla** |
> | -------------------- | ---------------- | :------: | :-----: | :--------------: | :--------: | :-------: | :---------: |
> | **Heat**    | Time (ms/sample) |  291.8  |  65.6  |      128.6      |   2770.5  |   361.3  |  65.2  |
> |                      | Peak Memory (GB) |   2.41   |   1.21  |       2.39       |    3.58    |    1.06   |   1.21  |
> | **Burgers** | Time (ms/sample) |  1371.0 |  780.6 |      211.2      |   8048.9  |   444.9  |  105.4 |
> |                      | Peak Memory (GB) |   2.57   |   1.26  |       2.43       |    3.65    |    1.08   |   1.22  |
> | **Reaction-Diffusion** | Time (ms/sample) | 636.6 | 744.4 | 159.9     |  6578.7 |   474.7  |  80.7 |
> |                      | Peak Memory (GB) |   2.97  |  1.72  |     2.93       |   4.33    |    1.43 |   1.43  |
> | **Navier–Stokes**    | Time (ms/sample) | 350.7 | 3355.6 | 678.8      | 16447.2   | 338.7  | 343.8 |
> |     | Peak Memory (GB) | 7.39  |   2.03    |  3.90     |  5.85  |  1.93   | 2.02 |
>
> ## R3-3: Generality Beyond Synthetic PDE Data
>
> We appreciate the reviewer’s suggestion and will clarify the intended scope of our generality claim. Specifically, by "general", we refer to PCFM’s ability to support **a wide variety of constraint types**—including **nonlinear**, **coupled**, and **global equality constraints**—on a **general class of widely studied PDE systems**. We chose flow matching models (FFMs) to reflect the functional nature of PDE data [1].
>
> To ensure comparability with prior work, we focused on **standard benchmark PDE datasets**, including **smooth equations** such as Heat and Reaction–Diffusion, and **nonlinear, shock-forming systems** such as Burgers and Navier–Stokes. These settings are commonly used in the constrained generative modeling literature \[2, 3, 4], neural operator learning \[5, 6], and foundation PDE models \[7, 8].
>
> We agree with the reviewer and will **soften claims about real-world generality** in the paper. While our method is broadly applicable to other constrained generation tasks (e.g., control, imaging, or molecular design), we reserve such extensions for future work.
>
> ## R3-4: Q1
>
> As per the reviewer's request, we also include an out-of-distribution (OOD) constraint experiment to assess PCFM's generalization beyond constraints implicitly present in the data during training.
>
> Specifically, we consider **initial condition (IC) shifts** in the **Burgers equation**, where the pretrained model has not seen such constraints during training. We generated a new test dataset where the location of the initial sigmoid profile $u(x,0) = \frac{1}{1 + \exp\left(\frac{x - p_{\text{loc}}}{\epsilon}\right)}$ is sampled from the intervals $[0.0, 0.1]$ and $[0.9, 1.0]$, while the training set used $p_{\text{loc}} \in [0.2, 0.8]$. The pretrained FFM model is **reused without retraining**, and all sampling methods are evaluated on the same OOD dataset to ensure fair comparison.
>
> We adopt 200 flow matching steps for the Burgers problem and the table below summarizes the results across methods.
>
> ### Table C: Performance on out-of-distribution dataset
>
> | Method       | MMSE       | SMSE       | CE (IC)     | CE (CL)     |
> | ------------ | ---------- | ---------- | ----------- | ----------- |
> | **PCFM**     | **0.0183** | **0.0140** | **0**       | **0**       |
> | ECI          | 0.1330     | 0.0565     | 0           | 3.6443e+00  |
> | DFlow        | 0.1559     | 0.0782     | 6.0027e+00  | 9.2038e-02  |
> | Vanilla      | 0.1884     | 0.0788     | 6.3685e+00  | 6.9078e-02  |
> | diffusionPDE | 0.1949     | 0.0803     | 6.4240e+00  | 6.2256e-02  |
> | PDM          | 2.0606     | 0.0281     | **0**       | 0           |
>
> PCFM achieves the lowest MMSE and SMSE while satisfying both ICs and nonlinear mass conservation even in this challenging OOD regime. These results demonstrate the **robust generalization** of our inference-time constraint enforcement, without requiring retraining or dataset-specific tuning. We will include this OOD evaluation in the final version of the paper.
>
> ## R3-4: Q2
>
> We thank the reviewer for the insightful question. As shown in Figure 9, increasing the number of flow matching steps improves MMSE by enabling the learned vector field to guide samples more accurately toward high-likelihood regions. However, we observe a mild increase in SMSE, particularly when stronger penalty weights are applied.
>
> This can be attributed to reduced sample diversity: more frequent projection-based corrections during integration may cause samples to collapse toward a narrower region of the constraint manifold, lowering ensemble spread. This behavior reflects a trade-off between sample fidelity (MMSE) and diversity (SMSE).
>
> Flow steps and step size are user-controlled hyperparameters in flow matching frameworks [9]. In our case, the flow integration length may slightly suppress SMSE due to stronger adherence to the constraint manifold. However, this is tunable and can be adapted based on application needs.
>
> [1] Kerrigan et al., arXiv:2305.17209, 2023.
>
> [2] Cheng et al., 2025.
>
> [3] Huang et al., arXiv:2406.17763, 2024.
>
> [4] Ben-Hamu et al., arXiv:2402.14017, 2024.
>
> [5] Li et al., arXiv:2010.08895, 2020.
>
> [6] Li et al., arXiv:2111.03794, 2021.
>
> [7] Herde et al., arXiv:2405.19101, 2024.
>
> [8] Sun et al., arXiv:2404.12355, 2024.
>
> [9] Lipman et al., arXiv:2412.06264, 2024.
>
> [10] Albergo et al., arXiv:2303.08797, 2023.

---

> > ### Comment · Reviewer_FfpW · 2025-08-02
> >
> > Let me begin by thanking the authors for their detailed response and inclusion of additional results during the rebuttal period. The explanations and supplementary experiments included in their response has addressed the major points in my review, and I am happy to reconsider my assessment in light of these:
> >
> > - **Comparison to ECI and PDM:** Thank you for this explanation! The additional analysis improves the positioning of PCFM within related literature, and I believe its inclusion will strengthen the paper. While I still maintain that these baseline methods closely resemble the methodological contributions provided in this study, which leaves remaining concerns regarding the novelty of the work, the inclusion of this additional empirical comparison in the final draft will strengthen the paper. In light of this, I believe the authors have addressed this concern as well as can be done during the rebuttal period, and I am willing to look past this point if the authors will include these explanations and results in an updated version of the paper.
> >
> > - **Runtime Analysis:** I appreciate the inclusion of these points in the response. While PCFM does incur additional overhead as compared to many of the baselines, this is not a significant concern -- it is OK to increase runtime in constrained settings such as those explored here. However, please include these in the paper as the table enhances transparency of considerations when adopting this approach.
> >
> > - **Generality Claims:** Thank you, this addresses my concern.
> >
> > - **Other Questions:** The OOD results are quite interesting, and I believe they have quite a bit of practical relevance to this area of the literature. Also, I appreciate the clarification of Q2 -- the explanation provided makes sense.
> >
> > In light of these clarifications, I will raise my score, with the expectations that these points will appear in the final version of the manuscript. Many thanks to the authors!

---

> ### Author Response · Authors · 2025-08-05
>
> Thank you for your thoughtful engagement and for reconsidering your assessment in light of our clarifications and additional results.
>
> We appreciate your acknowledgment of the empirical comparisons with ECI and PDM, and we agree that their inclusion in the main paper will help clarify the methodological distinctions and strengthen the positioning of PCFM. Nevertheless, PCFM employs a shoot–project–reverse framework with lightweight continuous guidance, which is inherently different from projecting intermediate states in the flow trajectory. This avoids PDM’s tendency to over-constrain noisy intermediates—especially under nonlinear constraints—and improves upon ECI, which is a special case of PCFM restricted to linear, decoupled constraints without penalty and lacks extensibility to nonlinear or global settings.
>
> We will incorporate the runtime and memory benchmarks into the main text to improve transparency regarding computational overhead. We also thank you for recognizing the practical relevance of the OOD experiments and the clarification regarding SMSE trends. As committed, all these analyses, comparison tables,  runtime metrics, and new baselines will appear in the final version.
>
> We are grateful for your constructive review and support. We also truly appreciate your decision to increase the score; it means a lot to us. Thank you!

---

### Official Review · Reviewer_aA1E · 2025-07-02

**Clarity:** 3
**Significance:** 2
**Originality:** 3
**Rating:** 4
**Confidence:** 4

**Summary:**

This paper proposes Physics-Constrained Flow Matching (PCFM), a zero-shot inference framework to enforce physical constraints  in flow-based generative models. PCFM corrects intermediate and final samples by projecting onto the constraint manifold, without requiring retraining or access to constraint gradients during training.

**Questions:**

See Weaknesses

**Ethical Concerns:**

["NO or VERY MINOR ethics concerns only"]

**Final Justification:**

I recommend this paper to accept when the authors promise to avoid using term "hard constraints" incorrectly.

**Limitations:**

See Weaknesses

**Quality:**

2

**Strengths And Weaknesses:**

Strengths:

- The shock fronts phenomenon when solving the Burgers's equation in Fig.1 are quite impressive.
- The paper identifies a core challenge in generative surrogate modeling for physics
- The writing is clear and easy to follow.

Weaknesses:

- I believe the term *hard constraint* may be an overstatement. In my view, a constraint can only be classified as *hard* if it is mathematically satisfied exactly (i.e., equals zero). Since your method merely reduces the violation to a small value rather than eliminating it entirely, it might be more appropriate to refer to it as a *soft constraint* or an *approximate hard constraint*.
- I am curious about the necessity of projecting the guess from intermediate flow states onto the constraint manifold. Given that physical constraints are typically imposed only on the final state, it is unclear what benefits such intermediate projections provide. Moreover, unless the constraint function h is affine, your projection step does not generally yield an exact solution on the target physics manifold. A more detailed justification would be helpful.
- In line 54, you mention that some methods require backpropagation through expensive PDE operators. However, your projection step also appears to be computationally intensive. It would be beneficial to include a comparison of computational time and memory consumption between your method and those baselines, to better contextualize the trade-offs involved.
- Could you provide additional qualitative comparisons between your method and all baseline models on the Navier-Stokes dataset? The other datasets used in your paper are relatively simple, which may limit the generalizability of your conclusions. Furthermore, experiments on higher-resolution or more complex datasets are highly recommended to more robustly demonstrate the effectiveness of your approach.

---

> ### Author Rebuttal · Authors · 2025-07-31
>
> We thank the reviewer for their constructive feedback and for appreciating the clarity of the writing, the motivation behind enforcing physical constraints in generative models, and the strength of the shock front results in Figure 1. Below, we address each of the key concerns in detail.
>
> ## R2-1: Use of the Term “Hard Constraint”
> We appreciate the reviewer’s attention to terminological precision. In our work, we enforce equality constraints via Gauss–Newton projection, with residual norms consistently reduced to within machine epsilon. In Float32, this corresponds to violations on the order of $10^{-7}$, and with Float64, the residuals drop to $\sim 10^{-15}$. This level of constraint satisfaction aligns with the definition of hard constraints in the numerical optimization literature, where constraints are considered “hard” if they are satisfied up to solver and machine precision [1].
> We will clarify this usage in the revised manuscript and explicitly state that our method achieves hard constraint satisfaction to the limits of numerical precision.
>
> ## R2-2: Necessity and Role of Intermediate Projections
>
> While constraints in physical systems (e.g., mass conservation or initial conditions) are often imposed only on the final state, we find that intermediate projections offer clear benefits for flow consistency and numerical stability. This aligns with practices in diffusion and flow matching literature, where intermediate states—often initialized as noise—are not expected to satisfy constraints or carry semantic meaning (e.g., intermediate samples in Stable Diffusion are noisy and not valid images).
>
> To assess this, we conducted an ablation in which projections were applied only at the final step, omitting intermediate corrections. Using the same setup (100 flow steps for Heat; 200 for Burgers, RD, and NS), we observed that trajectories drift farther from the constraint manifold, leading to larger corrective steps, slower convergence, and reduced accuracy. This confirms that relying solely on final correction is suboptimal, especially for nonlinear or global constraints.
>
> In contrast, intermediate projections act as **lightweight continuous guidance**, gradually steering samples toward the feasible set while preserving diversity. This avoids overly rigid enforcement on noisy intermediate states and improves final performance. Our results validate this approach, and we will include the ablation and clarify the motivation in the revision.
>
> ### Table A: Comparisons between only projecting final step and PCFM
> | Dataset | Method    | MMSE      | SMSE      | CE (IC) | CE (CL) |
> |---------|-----------|-----------|-----------|--------------|---------------|
> | Heat    | project final only  | 0.0431 |  0.0327 | 0 |  0 |
> | Heat    | PCFM      |    **0.00241**    |   **0.00937**       |          0    |        0       |
> | Burgers | project final only  | 0.1059 | 0.0690 | 0 | 4.0515 |
> | Burgers | PCFM      |   **0.00052**        |     **0.00272**      |      0      |      0         |
> | Reaction-Diffusion      | project final only  | 0.0282 | 0.0248 | 0 | 0 |
> | Reaction-Diffusion      | PCFM      |    **0.00026**       |     **0.00583**      |     0       |        0       |
> | Navier-Stokes      | project final only  | 0.1647 | 0.0860  | 0 | 0  |
> | Navier-Stokes      | PCFM      |    **0.0459**       |   **0.0417**       |    0          |      0         |
>
> ## R2-3: Computational Cost of Projection
>
> The reviewer correctly notes that constraint projections can introduce computational overhead. In PCFM, the projection step solves a Gauss–Newton system using iterative solvers (e.g., CG, GMRES) that require only Jacobian-vector products (Jv), computed efficiently via reverse-mode autodiff. We further accelerate this using batched vectorized maps (e.g., `vmap` in PyTorch) for GPU parallelism.
>
> Our implementation also supports **Jacobian-Free Newton–Krylov (JFNK)** methods, which approximate directional derivatives via finite differences, eliminating the need for explicit Jacobian construction and scaling well to high-dimensional and nonlocal constraints.
>
> For fair benchmarking, we use consistent batch sizes and 100 flow steps across all methods and datasets, evaluating average per-sample runtime and peak memory on a single NVIDIA V100 GPU. DFlow, which backpropagates through the entire ODE solve, is significantly slower and more memory-intensive; even with adjoint optimization, PCFM is 4–6× faster on average. In contrast, unconstrained baselines like vanilla FFM are faster but cannot guarantee constraint satisfaction. DiffusionPDE is faster than PCFM but sacrifices constraint enforcement and quality on nonlinear tasks. ECI’s runtime varies with mixing steps but fails to enforce nonlinear constraints robustly.
>
> Profiling (Table C) shows the final projection step contributes only 1–3% of total time; most cost is due to sampling itself. Overall, PCFM introduces a modest and practical overhead that is necessary to enforce hard nonlinear constraints, and we will summarize this analysis in the main paper.
>
> ### Table B: Runtime and Memory Comparison
>
> | **Dataset**          | **Metric**       | **PCFM** | **ECI** | **DiffusionPDE** | **D-Flow** |  **PDM**  | **Vanilla** |
> | -------------------- | ---------------- | :------: | :-----: | :--------------: | :--------: | :-------: | :---------: |
> | **Heat**    | Time (ms/sample) |  291.8  |  65.6  |      128.6      |   2770.5  |   361.3  |  65.2  |
> |                      | Peak Memory (GB) |   2.41   |   1.21  |       2.39       |    3.58    |    1.06   |   1.21  |
> | **Burgers** | Time (ms/sample) |  1371.0 |  780.6 |      211.2      |   8048.9  |   444.9  |  105.4 |
> |                      | Peak Memory (GB) |   2.57   |   1.26  |       2.43       |    3.65    |    1.08   |   1.22  |
> | **Reaction-Diffusion** | Time (ms/sample) | 636.6 | 744.4 | 159.9     |  6578.7 |   474.7  |  80.7 |
> |                      | Peak Memory (GB) |   2.97  |  1.72  |     2.93       |   4.33    |    1.43 |   1.43  |
> | **Navier–Stokes**    | Time (ms/sample) | 350.7 | 3355.6 | 678.8      | 16447.2   | 338.7  | 343.8 |
> |     | Peak Memory (GB) | 7.39  |   2.03    |  3.90     |  5.85  |  1.93   | 2.02 |
>
> ### Table C: PCFM Runtime Breakdown
>
> | **Dataset**          | **Component**                           | **Time (ms/sample)** | **Percentage** |
> | -------------------- | --------------------------------------- | ------------- | -------------- |
> | **Heat Equation**    | Sampling  | 286.6        | 98.22%         |
> |                      | Final Projection        | 5.19          | 1.78%          |
> |                      | **Total**                               | **291.8**    | 100.00%   |
> | **Burgers Equation** | Sampling | 1358.8        | 99.11%         |
> |                      | Final Projection         | 12.2          | 0.89%          |
> |                      | **Total**                               | **1371.0**    | 100.00%    |
> | **Reaction-Diffusion**    | Sampling |   630.2   |   98.99%       |
> |     | Final Projection   |    6.40     |       1.01%   |
> |        | **Total**                       | **636.6** | 100.00%    |
> | **Navier–Stokes**    | Sampling |    350.0    |     99.8%    |
> |                      | Final Projection    |  0.7   |     0.2%       |
> |                      | **Total**            |  **350.7**   |  100.00%    |
>
>
> ## R2-4: Generalizability and Experimental Scope
>
> We agree that demonstrating applicability beyond PDE-based systems would broaden the paper’s scope. Our primary focus is on scientific generative modeling, where physical constraints play a central role and quantitative evaluations are well-defined. To ensure comparability with prior work, we focused on **standard benchmark PDE datasets**, including **smooth equations** such as Heat and Reaction–Diffusion, and **nonlinear, shock-forming systems** such as Burgers and Navier–Stokes. These settings are commonly used in the constrained generative modeling literature \[2, 3, 4], neural operator learning \[5, 6], and foundation PDE models \[7, 8]. In addition, we also provide runtimes to all our datasets and add an additional baseline (PDM) \[9\], including on the three-dimensional Navier-Stokes dataset as well. The underlying FNO architecture in the PCFM can also be naturally adopted zero-shot super-resolution \[5\]. Nevertheless, we recognize the value of extending to more complex or real-world settings.
>
> [1] Nocedal, Jorge, and Stephen J. Wright. Numerical optimization. New York, NY: Springer New York, 2006.
>
> [2] Cheng, Chaoran, et al. "Gradient-free generation for hard-constrained systems." (2025).
>
> [3] Huang, Jiahe, et al. *DiffusionPDE: Generative PDE-solving under partial observation.* arXiv:2406.17763, 2024.
>
> [4] Ben-Hamu, Heli, et al. *D-flow: Differentiating through flows for controlled generation.* arXiv:2402.14017, 2024.
>
> [5] Li, Zongyi, et al. *Fourier neural operator for parametric partial differential equations.* arXiv:2010.08895, 2020.
>
> [6] Li, Zongyi, et al. *Physics-informed neural operator for learning partial differential equations.* arXiv:2111.03794, 2021.
>
> [7] Herde, Maximilian, et al. *Poseidon: Efficient Foundation Models for PDEs.* arXiv:2405.19101, 2024.
>
> [8] Sun, Jingmin, et al. *Towards a Foundation Model for Partial Differential Equation: Multi-Operator Learning and Extrapolation.* arXiv:2404.12355, 2024.
>
> [9] Christopher, Jacob K., Stephen Baek, and Nando Fioretto. "Constrained synthesis with projected diffusion models." Advances in Neural Information Processing Systems 37 (2024): 89307-89333.

---

> > ### Comment · Reviewer_aA1E · 2025-08-06
> > **Uncertain about "hard constraint"**
> >
> > Thank you for the clarification. I am still uncertain about your use of the term **“hard constraint.”** In numerical optimization I understand a hard constraint to require the residual to be **exactly zero**, not merely reduced to machine precision. Residuals on the order of $10^{-7}$ (single precision) or $10^{-15}$ (double precision) seem closer to **“numerically satisfied”** constraints, and it is not clear whether such accuracy can always be achieved for more complex equations.
> >
> > 1. Could you please indicate precisely where in reference \[1] this definition of “hard constraint” appears? I was unable to locate a clear statement in that book.
> > 2. If available, could you also provide additional, widely cited sources that explicitly classify constraints satisfied only up to machine precision as “hard”?
> >
> > Without such references, the current wording may overstate the strength of your guarantee.

---

> > > ### Author Response · Authors · 2025-08-06
> > >
> > > We thank the reviewer for the helpful follow-up and the opportunity to clarify our use of the term "hard constraint."
> > >
> > > **(1) Reference \[1] (Nocedal and Wright):**
> > > We acknowledge that Nocedal and Wright \[1] do not explicitly define the term “hard constraint.” However, Chapter 18 formulates equality-constrained problems using $h(x) = 0$, consistent with classical hard constraint formulations. In Algorithm 18.1, Sequential Quadratic Programming (SQP) is applied to enforce these constraints iteratively until convergence, where convergence is assessed via reductions in the constraint residuals. Appendix A further acknowledges that computations differ from exact arithmetic and that numerical methods aim to make these differences as small as possible, typically down to machine precision.
> > >
> > > Boyd and Vandenberghe \[2] similarly treat constraints as satisfied when residuals are reduced below a specified feasibility tolerance. For instance, Figure 11.21 shows that primal-dual residuals in an interior-point method converge to values near $10^{-15}$, which are treated as numerically zero ("residual converges rapidly to zero...") in double precision. This reflects standard practice in constrained optimization, where constraints are deemed satisfied when residuals fall below solver or machine precision.
> > >
> > > **(2) Use in recent ML and scientific computing literature:**
> > >
> > > In recent machine learning and scientific computing literature, the term “hard constraint” is consistently used to refer to constraints that are numerically enforced to within solver precision. For example, Donti et al. \[3] use differentiable constrained layers to ensure feasibility and refer to such constraints as hard. Négiar et al. \[4] enforce PDE constraints using a learned differentiable solver and describe satisfaction to machine precision as hard constraint enforcement. Lu et al. \[5] use augmented Lagrangian methods in physics-informed neural networks to reduce PDE residuals to $10^{-5}$–$10^{-8}$, which they describe as satisfying hard constraints.
> > >
> > > This usage aligns with the conventions adopted in widely used solvers such as IPOPT \[6] and MATLAB’s `fmincon` \[7], where equality and inequality constraints are considered satisfied when their residuals fall below a user-specified feasibility tolerance (e.g., $10^{-8}$ for IPOPT, $10^{-6}$ for `fmincon`). These thresholds are set to match the limits of floating-point precision and are widely accepted as sufficient for enforcing hard constraints in practice.
> > >
> > >
> > > While we agree that the constraint is not embedded by design into the model equations, our approach numerically enforces that such constraints are satisfied to the same accuracy as a hard constraint. In practice, there is no downstream equation or test that can distinguish our constraint satisfaction from exact enforcement within floating-point precision. while we agree that it could be called an "approximate hard constraint", such a constraint satisfaction is generally considered hard constraint satisfaction according to the fields such as Scientific Machine Learning \[4,5,8] and Differentiable Optimization \[3,6,7], and so we are simply choosing to go with the standard naming for clarity to readers rather than adopting a new scheme. We follow this established convention for clarity and consistency, and will revise the manuscript to explicitly state that “hard” refers to satisfaction up to solver and machine precision.
> > >
> > > \[1] Nocedal, J., & Wright, S. J. (2006). *Numerical Optimization* (2nd ed.). Springer.
> > >
> > > \[2] Boyd, S., & Vandenberghe, L. (2004). *Convex Optimization*. Cambridge University Press.
> > >
> > > \[3] Donti, P. L., Rolnick, D., & Kolter, J. Z. (2021). DC3: A Learning Method for Optimization with Hard Constraints. In *International Conference on Learning Representations (ICLR)*.
> > >
> > > \[4] Négiar, G., Mahoney, M. W., & Krishnapriyan, A. (2023). Learning Differentiable Solvers for Systems with Hard Constraints. In *The Eleventh International Conference on Learning Representations (ICLR)*.
> > >
> > > \[5] Lu, Lu, et al. "Physics-informed neural networks with hard constraints for inverse design." SIAM Journal on Scientific Computing 43.6 (2021): B1105-B1132.
> > >
> > > \[6] Wächter, A., & Biegler, L. T. (2006). On the Implementation of an Interior-Point Filter Line-Search Algorithm for Large-Scale Nonlinear Programming. *Mathematical Programming*, 106(1), 25–57.
> > >
> > > \[7] MATLAB Optimization Toolbox, `fmincon` Documentation. The MathWorks, Inc.
> > >
> > > \[8] Hansen, Derek, et al. "Learning physical models that can respect conservation laws." International Conference on Machine Learning. PMLR, 2023.

---

> > > > ### Comment · Reviewer_aA1E · 2025-08-07
> > > >
> > > > Thank you for the detailed clarification. However, I remain unconvinced that satisfying constraints up to machine precision alone justifies the term “hard constraint.”
> > > >
> > > > My concerns are as follows:
> > > > Nocedal & Wright [1] formulate equality‐constrained problems but nowhere equate “hard constraints” with “residuals below machine epsilon.”
> > > > Boyd & Vandenberghe [2, Fig. 11.21] merely illustrate convergence of residuals; they do not define such residual levels as “hard.”
> > > > Hence, the references do not appear to support the nomenclature.
> > > >
> > > > I think “approximately enforced” is a better choice and "hard constraint" is a over statement.

---

> ### Author Response · Authors · 2025-08-05
>
> Dear Reviewer aA1E,
>
> Thank you again for your thoughtful comments and questions, which have been very helpful in improving our paper. As the end of the discussion period approaches, we would like to gently remind you to consider our rebuttal and the updated version of the manuscript in your final evaluation.
>
> We hope that our responses have adequately addressed your concerns. If that is the case, we kindly ask you to consider raising your score to reflect this. Of course, if you have any remaining questions or would like further clarification, we would be happy to respond promptly.
>
> Thank you again for your time and engagement.
>
> Best regards,
>
> The authors

---

> ### Author Response · Authors · 2025-08-07
>
> Dear Reviewer aA1E,
>
> Thank you again for your follow-up. For elucidated clarity and in the spirit of aligning with your suggestion, **we will revise the manuscript to avoid the term "hard constraint" and instead adopt your recommended phrasing of "approximate hard constraint" or "approximately enforcing constraints", i.e., referring to constraint satisfaction up to machine precision.**
>
> We appreciate this distinction and agree that it more accurately reflects the practical guarantees achieved by our method. While we are effectively in agreement, we also wish to clarify that our original terminology was chosen to remain consistent with recent literature in scientific machine learning and optimization, where the term "hard constraint" is commonly used for constraints enforced up to numerical precision. As you requested earlier, we have also explicitly cited representative papers that adopt this convention, including recent works in scientific machine learning [1, 2, 3, 6] and widely used numerical optimization toolkits [4, 5].
>
> We hope this resolves the remaining concern. We would greatly appreciate your consideration in increasing the paper’s overall score in light of its contributions.
>
> Warm regards,
>
> The Authors
>
>
> [1] Donti, P. L., Rolnick, D., & Kolter, J. Z. (2021). DC3: A Learning Method for Optimization with Hard Constraints. In International Conference on Learning Representations (ICLR).
>
> [2] Négiar, G., Mahoney, M. W., & Krishnapriyan, A. (2023). Learning Differentiable Solvers for Systems with Hard Constraints. In The Eleventh International Conference on Learning Representations (ICLR).
>
> [3] Lu, Lu, et al. "Physics-informed neural networks with hard constraints for inverse design." SIAM Journal on Scientific Computing 43.6 (2021): B1105-B1132.
>
> [4] Wächter, A., & Biegler, L. T. (2006). On the Implementation of an Interior-Point Filter Line-Search Algorithm for Large-Scale Nonlinear Programming. Mathematical Programming, 106(1), 25–57.
>
> [5] MATLAB Optimization Toolbox, fmincon Documentation. The MathWorks, Inc.
>
> [6] Hansen, Derek, et al. "Learning physical models that can respect conservation laws." International Conference on Machine Learning. PMLR, 2023.

---

> > ### Comment · Reviewer_aA1E · 2025-08-07
> >
> > Thank you for your response. It solves my concern, and I will raise my score to 4.

---

> > > ### Author Response · Authors · 2025-08-08
> > >
> > > Thank you for your thoughtful review and for taking the time to consider our response. We truly appreciate your decision to raise the paper’s score, and your feedback has been valuable in helping us refine our work.

---

### Official Review · Reviewer_8V7Y · 2025-07-17

**Clarity:** 3
**Significance:** 3
**Originality:** 3
**Rating:** 5
**Confidence:** 1

**Summary:**

This work proposes Physics-Constrained Flow Matching (PCFM), a zero-shot inference framework for enforcing arbitrary nonlinear equality constraints in pretrained flow-based generative models. PCFM combines flow-based integration, tangent-space projection, and relaxed penalty corrections to strictly enforce constraints while staying consistent with the learned generative trajectory. This approach supports both local and global constraints, including conservation laws and consistency constraints, without requiring retraining or architectural changes. Empirically, PCFM outperforms state-of-the-art baselines across diverse PDEs, including systems with shocks and nonlinear dynamics, achieving lower error and exact constraint satisfaction.

**Questions:**

1.  A practical comparison of wall-clock inference times against the baselines would provide a more concrete sense of the computational efficiency of PCFM.

**Ethical Concerns:**

["NO or VERY MINOR ethics concerns only"]

**Final Justification:**

Thanks for the authors' rebuttal. My questions are clearly answered. I think the paper is technically solid with significant contribution. I keep the final rating unchanged.

**Paper Formatting Concerns:**

No.

**Quality:**

3

**Strengths And Weaknesses:**

I do not have enough background to review the paper. Here are a few comments:

**Strengths**

1. The "shoot-project-reverse" cycle designed in this work looks novel. This work presents an effective solution for satisfying the strict physical laws required when applying generative models to physical systems. It enforces complex, nonlinear equality constraints as hard requirements in a general way during the inference stage alone, without requiring to retrain the model or modify its architecture to introduce biases.

2. Empirical results in Table 3 looks promising. The proposed method seems much better than the existing counterparts.

**Weaknesses**

1.  In the step of forward shooting and projection, the cost depends on forming the constraint Jacobian, $J = \nabla h(u_1)$. For high-dimensional, complex, and non-local physical constraints, does constructing the Jacobian $J$ lead to a computational explosion?

---

> ### Author Rebuttal · Authors · 2025-07-31
>
> We thank the reviewer for their thoughtful assessment and for highlighting both the novelty of our shoot–project–reverse cycle and the empirical strength of our method across challenging physical domains. We are encouraged that the proposed framework was found effective and promising for enforcing strict physical constraints at inference time.
>
> ## R1.1: Computational Cost of Jacobian Construction
>
> We appreciate the reviewer raising this important question regarding scalability.
> In practice, the constraint Jacobian Jh(x) is not constructed explicitly. Instead, we compute it using batched reverse-mode automatic differentiation, enabled via vectorized mapping (e.g., vmap in PyTorch). This allows us to efficiently compute the full Jacobian across multiple constraint functions in parallel. Since each row of the Jacobian corresponds to a reverse-mode gradient computation, this formulation is highly parallelizable on GPUs and scales well even when the state dimension is large.
>
> For solving the projection step, employing iterative solvers (e.g., conjugate gradient, GMRES) on the Gauss–Newton system will require only Jacobian-vector products (Jv) rather than the full matrix. These products are efficiently handled by automatic differentiation and avoid the memory costs associated with storing large dense Jacobians.
>
> Additionally, our approach is compatible with Jacobian-Free Newton–Krylov (JFNK) methods, where even the Jv products are approximated via finite-difference evaluations. This further reduces memory footprint and makes the solver suitable for extremely high-dimensional or non-local constraint settings.
>
> In summary, the projection step does not pose a computational bottleneck, and we find it to be practical and scalable in our experiments.
>
> ## R1.2: Wall-clock inference times
>
> We provide additional results, measuring our inference times compared with the existing baselines. To compare fairly on time and peak memory, we use the same batch size for all methods and loop through the test set to record the per sample time and peak memory. We use 100 flow matching steps across all methods and datasets for benchmarking here. We use a single Nvidia V100 GPU with 32GB ram for all experiments. Batch sizes are 32 for heat, reaction diffusion, and Burgers problems and 8 for Navier-Stokes. We note that DFlow runs out of memory even for a batch size of 1 on the V100 chip, thus we improved the algorithm by including an adjoint ODE solve. For PDM, we adopt our constraint projection to every intermediate state along the flow matching trajectory. We report the average per-sample inference time and peak memory usage.
>
> PCFM incurs a modest runtime overhead compared to unconstrained baselines like vanilla FFM, due to projections to satisfy nonlinear hard constraints exactly. However, it achieves 4-6x speedups over D-Flow, which optimizes noise by backpropagating through the full ODE unroll - which is highly memory-intensive and slow. For Heat and Navier-Stokes datasets, the constraints are linear with well-defined structure, the projections converge in a single step, evident also in PDM's runtime. DiffusionPDE is faster than PCFM but it compromises on generative quality and struggles to enforce hard constraint, particularly for nonlinear problems like Burgers and Reaction-Diffusion. Similarly, ECI's mixing iterations can increase runtime overhead when tuned for more complex problems but it fails to achieve hard constraint satisfation for nonlinear problems.
>
> Profiling in Table B reveals 96-99% of PCFM's time is in intermediate projection steps, with final projection at 1-3% of the total runtime.
>
>
> ### Table A: Runtime and Memory Comparison
>
> | **Dataset**          | **Metric**       | **PCFM** | **ECI** | **DiffusionPDE** | **D-Flow** |  **PDM**  | **Vanilla** |
> | -------------------- | ---------------- | :------: | :-----: | :--------------: | :--------: | :-------: | :---------: |
> | **Heat**    | Time (ms/sample) |  291.8  |  65.6  |      128.6      |   2770.5  |   361.3  |  65.2  |
> |                      | Peak Memory (GB) |   2.41   |   1.21  |       2.39       |    3.58    |    1.06   |   1.21  |
> | **Burgers** | Time (ms/sample) |  1371.0 |  780.6 |      211.2      |   8048.9  |   444.9  |  105.4 |
> |                      | Peak Memory (GB) |   2.57   |   1.26  |       2.43       |    3.65    |    1.08   |   1.22  |
> | **Reaction-Diffusion** | Time (ms/sample) | 636.6 | 744.4 | 159.9     |  6578.7 |   474.7  |  80.7 |
> |                      | Peak Memory (GB) |   2.97  |  1.72  |     2.93       |   4.33    |    1.43 |   1.43  |
> | **Navier–Stokes**    | Time (ms/sample) | 350.7 | 3355.6 | 678.8      | 16447.2   | 338.7  | 343.8 |
> |     | Peak Memory (GB) | 7.39  |   2.03    |  3.90     |  5.85  |  1.93   | 2.02 |
> ---
>
>
> ### Table B: PCFM Runtime Breakdown
>
>
> | **Dataset**          | **Component**                           | **Time (ms/sample)** | **Percentage** |
> | -------------------- | --------------------------------------- | ------------- | -------------- |
> | **Heat Equation**    | Sampling  | 286.6        | 98.22%         |
> |                      | Final Projection        | 5.19          | 1.78%          |
> |                      | **Total**                               | **291.8**    | 100.00%   |
> | **Burgers Equation** | Sampling | 1358.8        | 99.11%         |
> |                      | Final Projection         | 12.2          | 0.89%          |
> |                      | **Total**                               | **1371.0**    | 100.00%    |
> | **Reaction-Diffusion**    | Sampling |   630.2   |   98.99%       |
> |     | Final Projection   |    6.40     |       1.01%   |
> |        | **Total**                       | **636.6** | 100.00%    |
> | **Navier–Stokes**    | Sampling |    350.0    |     99.8%    |
> |                      | Final Projection    |  0.7   |     0.2%       |
> |                      | **Total**            |  **350.7**   |  100.00%    |

---

> ### Author Response · Authors · 2025-08-05
>
> Dear Reviewer 8V7Y,
>
> Thank you again for your helpful comments and questions, which have contributed to improving our paper. As the end of the discussion period approaches, we would like to gently remind you to consider our rebuttal and the updated version of the manuscript in your final evaluation.
>
> We hope that we have satisfactorily addressed your concerns. If you have any further questions or would like additional clarification, we’d be happy to respond.
>
> Thank you once again for your time and consideration.
>
> Best regards,
>
> The Authors

---

### Note · Authors · 2025-08-12

**Dear Area Chair,**

As we approach the end of the rebuttal period, we would like to provide a concise summary of the discussion for important context in your decision. All four reviewers (8V7Y, aA1E, FfpW, and 5iKF) now recommend acceptance, with two maintaining clear accept scores and the other two increasing their ratings after discussion.

* **Reviewer 8V7Y** maintained their *Accept (5)* rating, finding the *shoot–project–reverse* cycle novel and effective for imposing complex constraints at inference time. We reached out after the rebuttal to check if any further clarifications were needed, but did not receive a response.

* **Reviewer aA1E** raised their score after we adopted their suggested phrasing of “approximate hard constraint,” which we will clarify further in the revised version, and after we provided an ablation demonstrating the benefit of intermediate projections. They also praised our results, noting that “the shock fronts phenomenon when solving the Burgers’s equation in Fig. 1 are quite impressive” and that “the paper identifies a core challenge in generative surrogate modeling for physics.”

* **Reviewer FfpW** also raised their score, acknowledging our added comparisons with ECI and PDM, the new out-of-distribution constraint experiment, and the expanded runtime analysis. These additions addressed the major concerns in their original review. They remarked that the OOD results are “quite interesting and practically relevant” and that runtime overhead is not a significant concern in constrained settings.

* **Reviewer 5iKF** maintained their *Accept (5)* rating, continuing to highlight PCFM’s ability to impose both linear and nonlinear constraints in a zero-shot manner. In response to our rebuttal, they agreed that “PCFM handles constraints—particularly nonlinear ones—more efficiently.”

We appreciate the reviewers’ engagement and the constructive discussion, which has led to broad agreement on PCFM’s technical merit, novelty, and practical relevance. We remain committed to incorporating all promised clarifications and additional results into the final version.

Warm regards,

The Authors

---

### Decision · Program_Chairs · 2025-09-17

**Decision:**

Accept (poster)

**Comment:**

The paper presents a method termed Physics-Constrained Flow Matching (PCFM), which enforces physical constraints in flow-based generative models via inference time guidance. This is implemented using the Gauss-Newton method.

The reviewers agreed that the paper is well written and that it contains significant results: the method is generally applicable and demonstrates strong empirical performance.

There were some concerns about the term "hard constraints" and about the novelty compared to ECI abd PDM but they were addressed in the rebuttal.

After the rebuttal, all reviewers recommend acceptance, provided that the authors update the paper according to the discussion (including relaxing the term "hard constraints").